# The urban carbon unlocking effect of digital infrastructure construction: A spatial difference-in-difference analysis from "Broadband China" pilot policy

Liang Guo[1], Lijing Chen[1]*, Zhen Yang[2,3]

**1** Academy of Fine Arts, Weifang University, Weifang, China, **2** School of Civil Engineering and Transportation, Weifang University, Weifang, China, **3** Innovation Center for CIM + Urban Regeneration, Qingdao University of Technology, Qingdao, China

* 20110913@wfu.edu.cn

**Data Availability Statement:** The data that support the findings of this study are openly available from the official website of the National Bureau of

## Abstract

As the foundation and cornerstone of the digital economy, digital infrastructure construction is an indispensable engine for realizing China's energy-saving and emission-reduction, innovation-driven and low-carbon transformation and development. Investigating the carbon unlocking effect of digital infrastructure construction might hasten the achievement of the dual-carbon goal and the "win-win" scenario of environmental protection and economic growth. However, there is still a gap between whether and how digital infrastructure construction can break the carbon lock-in (CLI). Based on the panel data of 266 prefecture-level cities from 2006 to 2019, this paper takes "Broadband China" policy (BCP) as a quasi-natural experiment, constructs a spatial difference-in-difference (SDID) model to explore its impact on CLI, and further analyzes its transmission mechanism, spatial spillover effect and heterogeneity. The results show that: (1) BCP can break the carbon lock-in in the pilot cities and remains valid after a series of robustness tests. (2) There is a lag in the carbon unlocking effect of BCP, and the effect is gradually significant after 3 years of policy implementation. (3) BCP has a spatial spillover impact on CLI, and it significantly contributes to both nearby and local cities. (4) By encouraging green technical advancement and upgrading industrial structure, BCP breaks the CLI. (5) There is regional heterogeneity and urban characteristic heterogeneity in the carbon unlocking effect of BCP. On this basis, we propose a series of policy recommendations to hasten the implementation of low-carbon transformation and sustainable urban development.

## 1. Introduction

Currently, human society is experiencing a digital social change based on broadband Internet information, and the digital revolution brought about by the spread of information technology has become an important factor in the economic growth of each country [1]. The digital wave

Statistics of China (Access: https://www.stats.gov.
cn/english/).

**Funding:** The author(s) received no specific
funding for this work.

**Competing interests:** The authors have declared
that no competing interests exist.

is driving the fourth industrial revolution in the world [2,3]. Internet information broadband
as a digital infrastructure has led to the rise of industrial Internet, cloud computing, artificial
intelligence and other technologies, which in turn promotes the green transformation of tradi-
tional industries, enables different innovation bodies to share and absorb knowledge with
higher efficiency and lower cost, and accelerates green technological innovation in the region
through the "spillover effect" [4–7]. In addition, the digital infrastructure realizes the linkage
of environmental information and resource sharing, making its role in environmental gover-
nance and supervision increasing prominent [8–10]. So, it is foreseeable that digital infrastruc-
ture provides a good opportunity for urban low-carbon transformation, industrial upgrading
and green development [11,12].

At the same time, many scholars believe that digital infrastructure has the potential to be
sustainable and are optimistic about reducing carbon emissions. Romm [13] found that wide-
spread use of the Internet resulted in a significant reduction in energy intensity in the United
States and highlighted the important contribution of ICTs in reducing greenhouse gases.
Moyer and Hughes [14] found that digital infrastructure could reduce global carbon emissions
by about 50 years. Liu and Zhang [15] verified that digital infrastructure can reduce carbon
emissions in China, and there is significant regional heterogeneity in this effect. Asongu Sim-
plice [16] found that digital infrastructure layout has a positive impact on reducing carbon
emissions in African countries. Zahra Dehghan Shabani [17] demonstrate that ICTs can
reduce carbon emissions in Iran's transport sector and countries along the Belt and Road. On
the one hand, digital infrastructure can optimize industrial structure and promote industrial
transformation and upgrading. The reduction of energy use intensity in the service sector
helps to reduce carbon emissions. On the other hand, digital infrastructure can promote tech-
nological progress, accelerate the research and development and diffusion of innovation, facili-
tate the development of clean energy, improve energy utilization efficiency, strengthen the
monitoring and control of carbon emissions, and thus reduce carbon emissions. It can be seen
that digital infrastructure helps to change the traditional development model from relying on
large-scale energy consumption to a more low-carbon and sustainable development mode,
effectively breaking the carbon lock of traditional industries, and providing key technical and
structural support for the global realization of a low-carbon economic path.

Therefore, a growing number of countries are proposing plans to grow their digital infra-
structure construction marked by broadband network, such as the United States, United King-
dom, and Japan [18–20]. In addition, the Chinese government has proposed a strategy known
as "Information Infrastructure Construction (IIC)". Specifically, since 2014, China's Ministry
of Industry and Information Technology has issued a "Broadband China" policy (BCP) and
has approved a total of three batches of pilot cities to promote information infrastructure con-
struction in 2015 and 2016. The policy mainly includes four contents in the upgrading of
broadband users in pilot cities, namely, scale, penetration rate, access capacity and application
scope. By then, China will have essentially finished building an accessible and fast internet net-
work infrastructure.

On the one hand, BCP, as a core component of digital infrastructure, provides the necessary
basic support for the development, services and applications of the digital economy by improv-
ing the quality and coverage of broadband networks. On the other hand, BCP is an important
national policy for the development of the digital economy, emphasizing the construction and
upgrading of digital infrastructure. The implementation of this policy has a direct impact on
the city's digitization process and related infrastructure investments. Therefore, we believe that
it is reasonable and feasible to take BCP as a proxy for digital infrastructure, and it is of great
significance. Our main objective is to focus on the phenomenon of CLI in the context of eco-
nomic development in modern countries, with a particular focus on the role and potential of

digital infrastructure development in addressing this problem. Our research results hope to find the positive impact of digital infrastructure construction represented by the Broadband network strategy to solve the CLI problem and provide practical experience and policy reference for other countries in the world facing similar problems.

Carbon lock-in (CLI) describes a phenomenon in which the fossil fuel-based energy consumption structure under the traditional economic development model cannot be changed in the short term, thus making the economy firmly locked into a carbon-based energy system [21–23]. CLI not only impedes the advancement of low-carbon technology and fosters dependency on development pathways, but also poses a persistent danger to harmony in ecosystems and environmental conservation[24]. Therefore, there is a pressing need to find a solution to the growing CLI, thereby decoupling economic growth from traditional development models.

Based on the above analysis, we infer that the BCP may have an impact on CLI. Moreover, China's ever-improving digital infrastructure is dramatically changing its development model. In this regard, China's digital transformation and commitment to the "dual carbon" goal make it a good practical case to study the BCP-CLI relationship. Therefore, we cannot help but ask: (1) How does BCP break the China's CLI? (2) Does BCP have spatial spillover effects and heterogeneity on CLI? The motivation of this study is to answer the above questions and investigate to what extent BCP can influence CLI.

The following are some contributions: First, we constructed an indicator evaluation system for CLI at the city level and measured and analyzed it spatially and temporally. Second, we integrate BCP and CLI into the same theoretical analysis framework for the first time and explore the impact and mechanism of BCP on CLI. This helps accelerate the urban carbon neutrality process and provides decision support for solving the dilemma of urban economic development and environmental protection. Furthermore, this study reveals the spatial spillover effect of BCP on CLI, providing a new strategy for regional synergistic emission reduction and environmental governance. Finally, this study explores the heterogeneous effects of BCP on CLI, which will help the central government to continuously improve the relevant policies and develop a more precise and effective CLI multi-governance system in a targeted manner.

The rest of the paper is structured as follows: Section 2 summarizes the findings of the literature and provides the theoretical hypotheses; Section 3 outlines the data and methods; the research findings are summarized in section 4; Section 5 goes into great depth about the mediating effects and heterogeneity between BCP and CLI, and the last section concludes with policy proposals.

## 2. Literature review and research hypothesis

### 2.1 Introduction of CLI

CLI is characterized by stability and reinforcement, embedding economic growth in a continuing development model that depends on crudely exploiting conventional energy and resources [25,26]. The most typical examples of CLI are some infrastructures, such as coal power plants, iron plants, steel plants, thermal power plants, and parts of the transportation infrastructure [27,28]. Investments in such carbon-intensive and energy-intensive infrastructures inevitably lead to CLI, as these infrastructures have an extremely long lifespan and generate carbon emissions directly and indirectly from their installation and operation [25]. The World Resources Institute (WRI) estimates that infrastructure and equipment have an average lifespan of 27.5 years, but coal plants lasting even longer than 40 years, and their carbon emissions over the entire life cycle are 68 times that of nuclear power plants. The majority of academics concur that when the technology-institution combination is established, the interactions between

firms, individuals, and governments will continue to reinforce CLI after undergoing market and social integration [29].

## 2.2 Research about CLI

CLI is a problem that China and other countries must face in the process of realizing their carbon reduction targets [30]. However, considering that the concept of technology-institution complex has not yet been clearly defined, scholars have interpreted it from the technological level [31], industrial level [32], and the regional level [33], respectively, according to their own understanding. CLI is directly exacerbated by carbon-based technologies [34]. The interaction of technological lock-in, institutional lock-in and consumption lock-in deepens the degree of CLI [25,27]. Currently, research on CLI focuses on two aspects, with one part of scholars attempting to assess CLI and the others exploring the influencing factors affecting CLI.

Despite the gradual deepening research on CLI in recent years, there is still no standardized assessment method. Dong and Li [35] calculated China's CLI based on the process of carbon sinks and carbon emissions. Other scholars have constructed a comprehensive indicator system to measure CLI [26,36–38]. Their attention is undoubtedly focused on two points, one is technological lock-in and the other is institutional lock-in, which is the core of CLI that we mentioned in the previous section.

A number of scholars have examined the influences on CLI and the relationship with other socioeconomic factors. Based on the viewpoint of industrial transfer, Xu and Dong [39] discovered that technical advancement might undermine the CLI both directly and indirectly. Driscoll [23] argues that government-sponsored policies could greatly impact carbon emissions, and it has been discovered that transportation infrastructure measures can dramatically reduce CLI in the transportation sector. According to Oberthür and Khandekar [40], global governance can help states and transnational organizations reduce the CLI. In addition, other aspects of CLI, including electricity [41], finance [29,42], social welfare [43,44] and innovation [36,38,45] have also been analyzed.

## 2.3 Investigation into the relationship between digital development and carbon emissions

Numerous studies have looked at how digital technologies affect carbon emissions to date [12,46–48], but there is still some controversy. For example, Sadorsky [49] argues that information and communications technology (ICT) development can raise power usage and thus carbon emissions. This conclusion is supported by the investigations of Collard and Fève, who find that ICT do not contribute to improved energy usage and efficiency [50]. Salahuddin and Alam [51] also fails to identify any benefits of ICT for energy using panel data from OECD countries. Khan [52], based on data from N-11 countries, found that the use of ICT devices actually increased carbon emissions. However, Lu [53]found a positive carbon mitigation effect of ICT in 12 Asian countries and considered it as an important strategy to achieve low carbon development. Raheem and Tiwari [54] demonstrated the G7 countries' ability to reduce carbon emissions in the long run thanks to ICT. Škare and Gavurova [55] examines the impact of digitalization on the carbon footprint of governments, households, businesses, NGOs and imports across the EU and draws positive conclusions. Bocean [56] assesses the positive impact of digital transformation on the economic performance and sustainability of EU countries. Based on data at a global level, Zuo and Zhan [57] found that the impact of digital development on carbon reduction is much stronger in Europe and North America than in other countries. In addition, a few scholars believe that the impact of ICT on carbon emissions is not significant. Based on Tunisia's long time series data, Amri and Zaied [58] found that

ICT has negligible impact on carbon emissions, which means that the Tunisian government can promote the development of ICT.

Meanwhile, other academics have investigated how digital technology affects carbon emissions in certain regions. For example, Shahnazi and shabani [59] found that the popularization of digital technology played an important role in carbon emission reduction in Iran. Gay and Davis [60] used e-commerce data to find that digital technology had a positive effect on the environmental impacts in United States, especially on carbon emissions. However, Amri [61] finds that digital technology has no effect on carbon emission reduction in Tunisia. As for China, several scholarly works have carefully examined the connection between digital technology and carbon emissions. For instance, Hu and Zhang [11] discover that industrial restructuring and digital infrastructure can help Chinese cities turn into low-carbon cities, and Dong and Yang [62] find that through structural and technical consequences, digitalization may lower carbon emissions. Yang and Gao [3] construct a comprehensive indicator system of digitalization and finds that the carbon emission intensity is greatly reduced during the digital city construction. Additionally, several researchers have looked at how spatially specific digital technologies affect the environment. For example, Su and Li [63] point out that significant spatial spillover effects are caused by digital financial technologies, and Hao and Peng [64] find that regional carbon emissions spread and spill over more widely due to the geographic influence between cities.

It is not difficult to conclude that there is essentially no literature relating to BCP and CLI. Despite the fact that some academics have examined how digital technology, digital infrastructure, and digital transformation policies affect carbon emissions, the BCP-CLI link remains a research gap. Moreover, the long-term effects of BCP on CLI are equally worth analyzing, as the impacts of these policies can be long-lasting. However, empirical analysis, mechanisms and heterogeneity have not been the focus of much research.

### 2.4 Research hypotheses

The continuous improvement of digital technology has laid the foundation for the city's digital transformation, which can enhance the level of intelligent manufacturing and promote green and low-carbon development. For example, smart manufacturing workshops can realize the control of carbon emissions in data integration, prediction and analysis [65]. In addition, digital technology can establish a smart energy management system for enterprises to reduce energy consumption and improve energy efficiency by optimizing the production process [66,67]. Digital technology accelerates the flow of innovative elements, reduces the spatial barriers of information transmission [68], strengthens the application of innovative technologies [69], effectively promotes the circular economy and industrial structure optimization [70], thus solving the technological lock-in and industrial lock-in. The openness, interactivity and effectiveness of the digital network make up for the deficiencies of the government in environmental management and promote the modernization of environmental governance [70,71]. Digital technologies have also increased the public's avenues for involvement in environmental governance and given non-governmental groups a role in environmental oversight, which has resulted in encouraged green manufacturing practices [72]. In addition, digital technology has given birth to the intelligence of social life. The home office model reduces energy consumption, and intelligent logistics improves supply chain's efficiency, thus solving institutional lock and social lock [73]. In light of this, we propose the following hypothesis:

**Hypothesis 1.** BCP can break the CLI.

Digital technology plays a pivotal role in regional innovation [74]. First, digital technology enables a high degree of integration of information after processing, thus optimizing the whole

innovation activities process [75]. Second, the spread of knowledge and information has hastened because to the growth of digital technology, which has also increased the innovation efficiency [76]. Third, instant messaging and telecommuting were produced from digital technology, have altered how people work and live [77]. Fourth, digital technologies provide the "raw material" for regional innovation and can accelerate innovation output [78]. To a certain extent, this lowers transaction costs, reduces resource consumption and increases productivity, which can lead to improved carbon emission performance and thus break the CLI.

The integration of digital technology to the whole process of products accelerates the renewal of the product cycle, which can encourage the modernization and change of the industrial structure [73]. The continual advancement of digital technology has led to the steady emergence of novel business models and other forms of economic growth, supporting the transformation and modernization of industrial structure [79]. For example, the intelligent transformation of the service industry can accelerate the formation of a new industrial production service system and realize the transformation of the traditional consumption structure into green and low-carbon consumption. Second, The factor structure is optimized by the integration of conventional industry and digital technologies [80]. Third, new technologies integrated into established industries lessen reliance on natural resources, enhance the labor force distribution, thus support the modernization of industrial institutions [72,74]. The proportion of traditional intensive industries declines, and new types of business continue to emerge, thus realizing the improvement of carbon emission efficiency and thus breaking the CLI. In light of this, we propose the following hypothesis:

**Hypothesis 2.** BCP can break the CLI through technological innovation and industrial structure upgrading.

BCP facilitates the transfer of green technologies, ideas and institutions to neighboring regions [5]. Digital technologies can spread to the neighborhood through shared networks and platforms, creating a "spillover effect" that breaks the CLI [81]. Also, digital technologies enable the public to access information from multiple sources and raise their environmental awareness, thus breaking the CLI [82]. The combination of digital technology and government governance has led local governments at all levels to compete to put in place digital governance regulations and systems [54] in order to avoid being eliminated from the new environmentally oriented regional race. Therefore, if the policy is effective, other regions are bound to follow suit. In addition, regions with higher digital technology are more inclined to attract capital intervention, while regions with scarce digital technology will still retain backward productivity, thus reinforcing CLI [83]. This situation also exists in China. Over the past decade or so, there have been serious regional disparities in digital infrastructure [84], which in turn has been an important factor in attracting talent and capital [85]. Thus, the development of BCP can affect CLI through the "siphon effect". In light of this, we propose the following hypothesis:

**Hypothesis 3.** BCP has a spatial spillover effect on CLI.

Based on the above analysis, we mapped the mechanism by which BCP affects CLI, as shown in Fig 1. In addition, we have also drawn a methodological flow chart for this study, as shown in Fig 2.

## 3. Method and data

### 3.1 Model setting

We view BCP as a quasi-natural experiment, and the best tool for evaluating the impact of policy is the DID method. However, using the traditional DID method alone to measure the

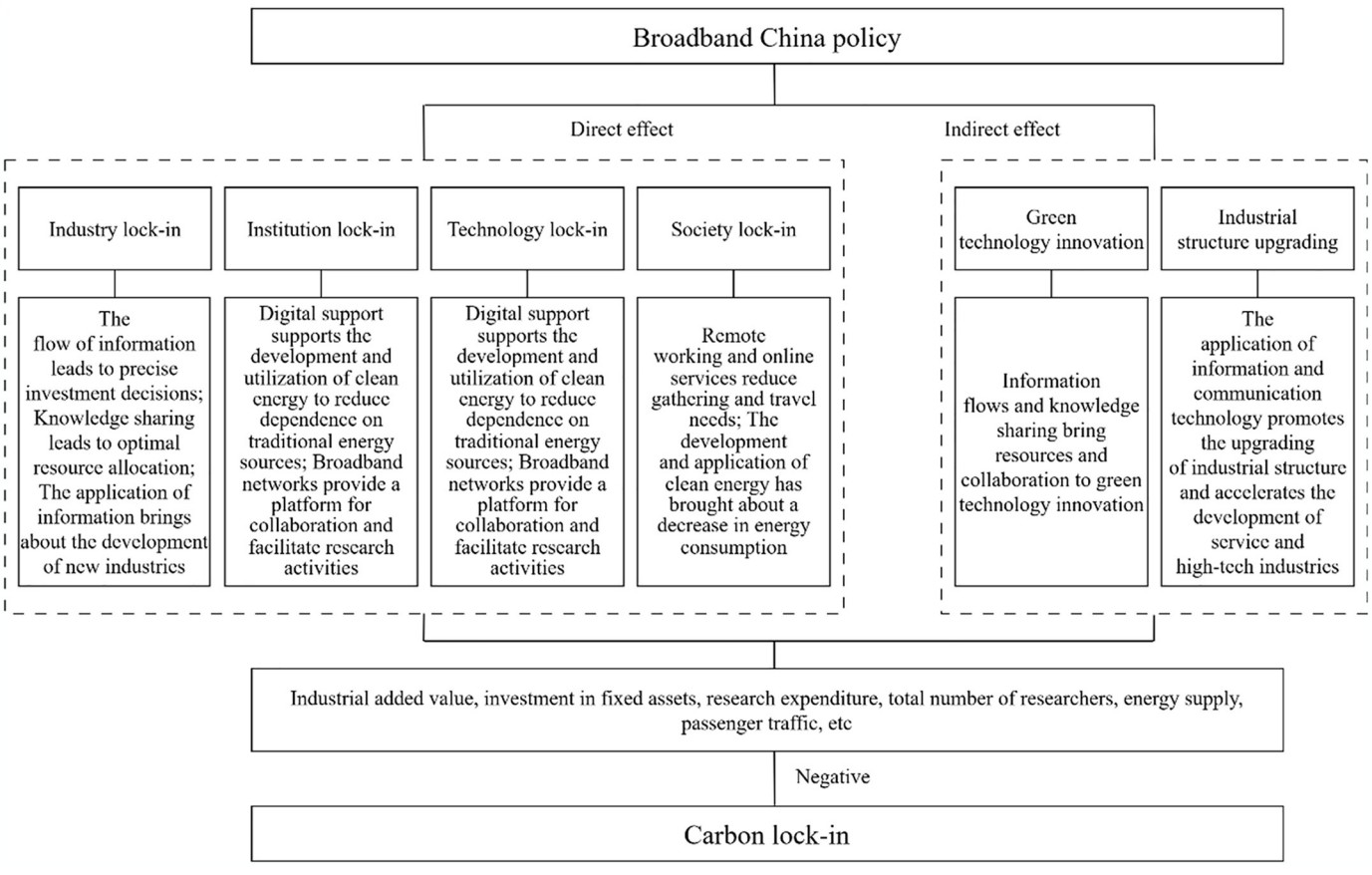

**Fig 1. The impact mechanisms of the BCP on CLI.**

impact of BCP on the CLI is somewhat biased, because there may be some spatial effects. Therefore, this paper applied the SDID approach to investigate the mechanisms and spatial effects of BCP affects CLI, which has also been widely adopted in the literature and has yielded robust results [38,86,87]. In particular, three forms of spatial econometric models exist depending on the object of study, i.e., spatial error model (SEM), spatial lagged model (SLM), and spatial Durbin model (SDM). The general expression formulas for the above three econometric models are as follows:

$$CLI_{it} = \delta W_\theta CLI_{it} + \varphi_0 + \sum \tau_i X_{it} + \sum \gamma_m W_\theta X_{it} + \pi_i + \mu_t + \varepsilon_{it} \qquad (1)$$

where $CLI_{it}$ is the explanatory variable, i.e., the CLI index. $X_{it}$ is the independent variable, i.e., whether the broadband China pilot policy is implemented or not. $W_\theta$ is the spatial weight matrix of n*n, and n is the number of cities. $W_\theta X_{it}$ represents the spatial lag term of the independent variable, and $W_\theta CLI_{it}$ represents the spatial lag term of the dependent variable. $\varphi$, $\delta$, and $\tau$ denote the marginal effect of the explanatory variable, the marginal effect of the dependent variable, and the marginal spatial effect of the explanatory variable, respectively. In addition, $\pi_i$, $\mu_t$, and $\varepsilon_{it}$ represent the individual effect, time effect, and random error term, respectively. It should be especially noted that SDM is able to degenerate into SEM or SLM under special circumstances, so it is necessary to combine different test results to determine which model is the most appropriate to use in practical applications.

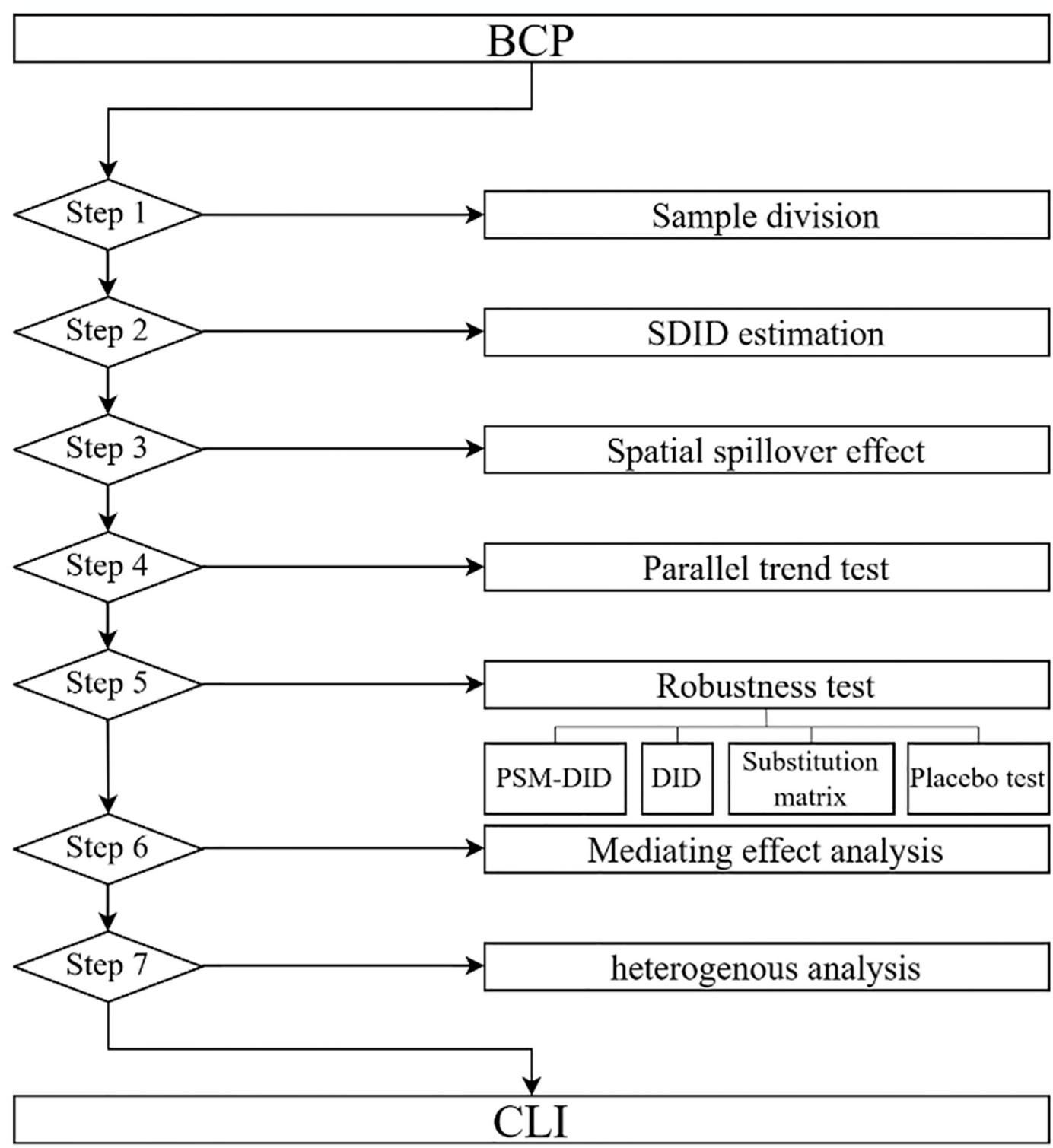

**Fig 2. Methodological flow.**

Therefore, this study uses CLI as the dependent variable and BCP pilot policy implementation as the explanatory variable, with a number of control factors taken into account to prevent the possibility of estimate bias. The specific expressions of SDM, SEM and SLM are as follows:

$$CLI_{it} = \delta W_\theta CLI_{it} + \varphi_0 + \varphi_1 BCP_{it} + \varphi_2 W_\theta BCP_{it} + \sum \tau_i X_{it} + \sum \gamma_m W_\theta X_{it} + \pi_i + \mu_t + \varepsilon_{it} \quad (2)$$

$$\begin{cases} CLI_{it} = \varphi_0 + \varphi_1 BCP_{it} + \sum \tau_i X_{it} + +\pi_i + \mu_t + \varepsilon_{it} \\ \varepsilon_{it} = \alpha W_\theta \varepsilon_{it} + \vartheta \end{cases} \quad (3)$$

$$CLI_{it} = \delta W_\theta CLI_{it} + \varphi_0 + \varphi_1 BCP_{it} + \sum \gamma_m W_\theta X_{it} + \pi_i + \mu_t + \varepsilon_{it} \quad (4)$$

## 3.2 Variables and data

In this paper, the dependent variable is the BCP city pilot policy, which is the interaction term between the pilot city variable and the time variable. Specifically, $BCP_{it} = 1$ if a city has implemented a BCP city pilot at a certain time, and 0 otherwise. The data presented here were taken from the State Council of the Chinese Central People's Government's Policy Documents Library.

The dependent variable in this paper is CLI at the city level. As mentioned earlier, although research on CLI is gradually deepening, there is still no standard of measurement. A part of scholars adopts the indicators of carbon sinks and carbon emissions to measure CLI, but disregard the two most crucial implications, i.e., institution and technology. Therefore, in this paper, drawing on the methods proposed by Niu and Liu (26) Niu, Liu [26] and Zhao, Taghizadeh-Hesary [29], we comprehensively quantify the CLI at the city level from four aspects, namely, technology, institution, society and industry, and construct the indicator system as shown in Table 1. Importantly, the metrics of the CLI we built are consistent with the characteristics of the BCP. First, broadband construction promotes the development of the information technology industry, which will affect the structural changes of the secondary industry, especially by improving production efficiency and creating new business models to guide the direction of fixed asset investment, and then affect the industry lock-in. Second, the implementation of BCP will affect the local science and technology expenditure and talent structure,

**Table 1. Index system for measuring CLI in China.**

| Indicator | Introduction | Property |
|---|---|---|
| Industry lock-in | The proportion of added value of the secondary industry to GDP | Positive |
| | Proportion of fixed assets investment in GDP | Positive |
| | The proportion of industrial added value to GDP | Positive |
| Institutional lock-in | Total employment in the mining industry | Positive |
| | Local science and technology expenditure | Negative |
| | Number of scientific research personnel in enterprises and institutions | Negative |
| Technological lock-in | Energy intensity | Positive |
| | The proportion of investment in research and development expenditure to GDP | Negative |
| | Carbon intensity | Positive |
| Social lock-in | Population density | Positive |
| | Total natural gas supply | Negative |
| | Total passenger volume | Positive |

promote local governments to increase investment in science and technology research and development and talent training, and then affect the institutional lock-in. Moreover, the increase in broadband may facilitate the rapid adoption of new technologies (such as cloud computing, big data analytics, etc.) and reduce energy intensity and carbon intensity, thereby affecting technological lock-in. Finally, BCP can improve the accessibility of information and drive changes in social behavior. For example, better connectivity may facilitate remote working and online services, thereby reducing reliance on traditional modes of transport, affecting total ridership and, in turn, social lock-in.

Considering that other factors may affect the CLI, this paper controls for the following five variables based on reference to other literature: (1) economic development conditions (pgdp), which is expressed by GDP per capita; (2) level of openness (fdi), which is calculated by the total of foreign direct investment; (3) human capital (stu), which is expressed by the total number of university students enrolled in the university; (4) government governance capacity (fin), which is expressed by the total general budget revenue of local finance; (5) openness (ope), which is expressed by the total exports and imports. In addition, in order to explore the mechanism by which BCP affects CLI, this paper constructs two mediating variables, namely, technological innovation (tec) and industrial structure (ins), the former is expressed by the total amount of green invention patents of prefecture-level cities in the year, and the latter is evaluated by the percentage of the added value of the tertiary industry to GDP.

The second and third rounds of the BCP pilot, which has been in operation since 2014, were carried out in 2015 and 2016, respectively, with a cumulative total of 108 prefectural-level cities successively selected (in this paper, cities with unavailable data are excluded). Therefore, we considered the first eight years of policy enforcement and the three years after policy enforcement, i.e., 2006–2019, as the sample intervals. The carbon emission data was measured using the method of Wu and Guo [88]. The data used in other indicators, including those in the control variables, are obtained from the China Statistical Yearbook, the China Urban Yearbook, and the national economic and social development bulletins of each city, and missing data are filled in using the interpolation method. Finally, we obtain 14-year balanced panel data that include 266 prefecture-level cities, totaling 3724 observations. Please refer to Table 2 for more details on these variables.

## 4. Empirical results

### 4.1 Descriptive analysis of CLI

We adopted principal component analysis to calculate the CLI levels at the national and prefecture separately from 2006–2019. Fig 3 depicts the trend of CLI at the national level. Table 3

**Table 2. Descriptive statistics for all variables.**

| Variables | Obs | Mean | Std. Dev. | Min | Max |
|---|---|---|---|---|---|
| did | 3724 | .142 | .349 | 0 | 1 |
| lncli | 3724 | -2.095 | .463 | -3.517 | -.454 |
| lnpgdp | 3724 | 10.441 | .705 | 8.249 | 12.324 |
| lnfdi | 3724 | 9.822 | 1.962 | .693 | 14.941 |
| lnstu | 3724 | 10.453 | 1.426 | 4.234 | 13.897 |
| lnfin | 3724 | 13.616 | 1.211 | 9.722 | 18.087 |
| lnope | 3724 | 13.944 | 2.168 | 3.526 | 19.658 |
| lntec | 3724 | 4.351 | 1.948 | 1 | 10.88 |
| lnins | 3724 | 3.638 | .258 | 2.149 | 4.425 |

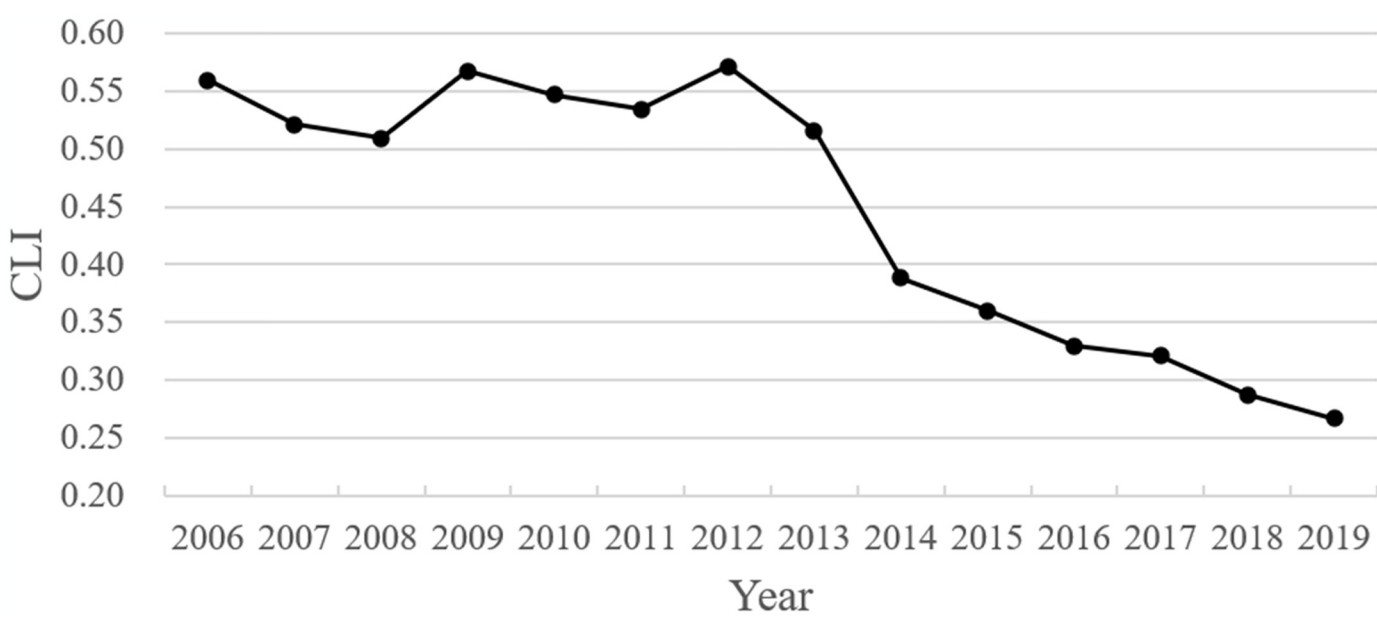

**Fig 3. Trends in CLI changes at the national level from 2006 to 2019.**

employed the natural breaks method to categorize the CLI of prefecture-level cities and summarized the cities with the 10 highest CLI values in each category separately.

## 4.2 Benchmark regression results

Table 4 summarizes the results of the spatial correlation test for CLI from 2006–2019, and Fig 4 plots the associated Moran scatterplot. From Table 4 and Fig 4, it can be found that there is a significant positive correlation for CLI, which suggests that investigating the association between BCP and CLI requires the use of spatial econometric models. After the Lagrange multiplier (LM) test, Hausman test, Wald test, and likelihood ratio (LR) test, the above results support our use of the two-way fixed effects (individual and time) of the SDM model as the benchmark regression model.

The outcomes in column (5) of Table 5 show that the coefficient of BCP is significantly negative, which indicates that digital development is able to break the CLI. Specifically, compared to non-pilot cities, BCP pilot cities are able to reduce the local CLI by 2.54%. Therefore, hypothesis 1 is verified.

## 4.3 Spatial spillover effects of BCP on CLI

As mentioned before, there is a spatial effect of BCP affecting CLI, so to determine the spatial spillover effects of BCP on CLI, the partial differentiation approach need to be performed [89], and the outcomes are shown in Table 6. From Table 6, it can be found that the BCP is able to reduce the CLI by 3.19%, a finding that is in line with the benchmark regression outcomes, indicating that BCP is effective and can significantly reduce the local CLI. In addition, all other control variables are in line with the benchmark regression model. Specifically, economic development is conducive to curbing the local CLI, while human capital, fixed asset investment, openness, and government governance exacerbate the local CLI. Moreover, Table 5 reports the indirect effect of BCP affecting CLI. That is, there is an inhibitory effect of the BCP on CLI of neighboring cities. From the regression coefficients, the indirect effect of BCP is

**Table 3. CLI classification of prefecture-level cities in 2006, 2010, 2015 and 2019.**

| Year | CLI categories | Cities |
|---|---|---|
| 2006 | 0.056–0.104 | Tangshan, Handan, Datong, Yangquan, Jincheng, Daqing, Xuzhou, Huainan, Huaibei, Zaozhuang |
| | 0.105–0.143 | Qinhuangdao, Baoding, Zhangjiakou, Chengde, Langfang, Hengshui, Yuncheng, Tongliao, Dalian, Anshan |
| | 0.144–0.207 | Beijing, Shijiazhuang, Xingtai, Cangzhou, Shuozhou, Xinzhou, Baotou, Chifeng, Ordos, Shenyang |
| | 0.208–0.311 | Tianjin, Taiyuan, Changzhi, Jinzhong, Linfen, Lvliang, Hulunbuir, Fuxin, Panjin, Tieling |
| | 0.312–0.471 | Tangshan, Handan, Datong, Yangquan, Jincheng, Daqing, Xuzhou, Huainan, Huaibei, Zaozhuang |
| 2010 | 0.067–0.115 | Qinhuangdao, Hengshui, Hohhot, Dalian, Anshan, Dandong, Jinzhou, Siping, Harbin, Qiqihar |
| | 0.116–0.157 | Shijiazhuang, Baoding, Chengde, Langfang, Yuncheng, Baotou, Tongliao, Ordos, Yingkou, Liaoyang |
| | 0.158–0.219 | Xingtai, Zhangjiakou, Cangzhou, Shuozhou, Xinzhou, Chifeng, Hulunbuir, Shenyang, Fushun, Huludao |
| | 0.220–0.313 | Handan, Taiyuan, Changzhi, Jinzhong, Linfen, Lvliang, Fuxin, Jixi, Hegang, Qitaihe |
| | 0.314–0.493 | Beijing, Tianjin, Tangshan, Datong, Yangquan, Jincheng, Panjin, Daqing, Huainan, Huaibei |
| 2015 | 0.064–0.118 | Chengde, Hohhot, Tongliao, Ordos, Dalian, Dandong, Jinzhou, Huludao, Siping, Tonghua |
| | 0.119–0.166 | Shijiazhuang, Qinhuangdao, Baoding, Zhangjiakou, Cangzhou, Langfang, Hengshui, Yuncheng, Baotou, Chifeng |
| | 0.167–0.247 | Xingtai, Shuozhou, Xinzhou, Hulunbuir, Shenyang, Fushun, Fuxin, Tieling, Songyuan, Jixi |
| | 0.248–0.360 | Beijing, Tianjin, Tangshan, Handan, Taiyuan, Jinzhong, Linfen, Lvliang, Panjin, Daqing |
| | 0.361–0.599 | Datong, Yangquan, Jincheng, Jining, Pingdingshan, Chongqing |
| 2019 | 0.030–0.064 | Qinhuangdao, Zhangjiakou, Chengde, Hengshui, Hohhot, Dalian, Dandong, Jinzhou, Yingkou, Fuxin |
| | 0.065–0.094 | Shijiazhuang, Baoding, Yuncheng, Baotou, Chifeng, Anshan, Benxi, Liaoyang, Huludao, Changchun |
| | 0.095–0.151 | Handan, Xingtai, Cangzhou, Langfang, Shuozhou, Xinzhou, Hulunbuir, Shenyang, Fushun, Tieling |
| | 0.152–0.265 | Beijing, Tianjin, Tangshan, Taiyuan, Datong, Yangquan, Jinzhong, Linfen, Lvliang, Ordos |
| | 0.266–0.505 | Changzhi, Jincheng |

**Table 4. Global spatial autocorrelation test results of CLI.**

| Year | Moran'I | z-score | p-value |
|---|---|---|---|
| 2006 | 0.067 | 14.475 | 0.000 |
| 2007 | 0.076 | 12.735 | 0.000 |
| 2008 | 0.080 | 15.076 | 0.000 |
| 2009 | 0.080 | 15.193 | 0.000 |
| 2010 | 0.077 | 14.554 | 0.001 |
| 2011 | 0.073 | 13.850 | 0.000 |
| 2012 | 0.074 | 13.387 | 0.000 |
| 2013 | 0.070 | 14.078 | 0.000 |
| 2014 | 0.082 | 15.489 | 0.000 |
| 2015 | 0.075 | 14.313 | 0.001 |
| 2016 | 0.098 | 18.320 | 0.000 |
| 2017 | 0.095 | 17.800 | 0.000 |
| 2018 | 0.084 | 15.907 | 0.000 |
| 2019 | 0.092 | 17.407 | 0.001 |

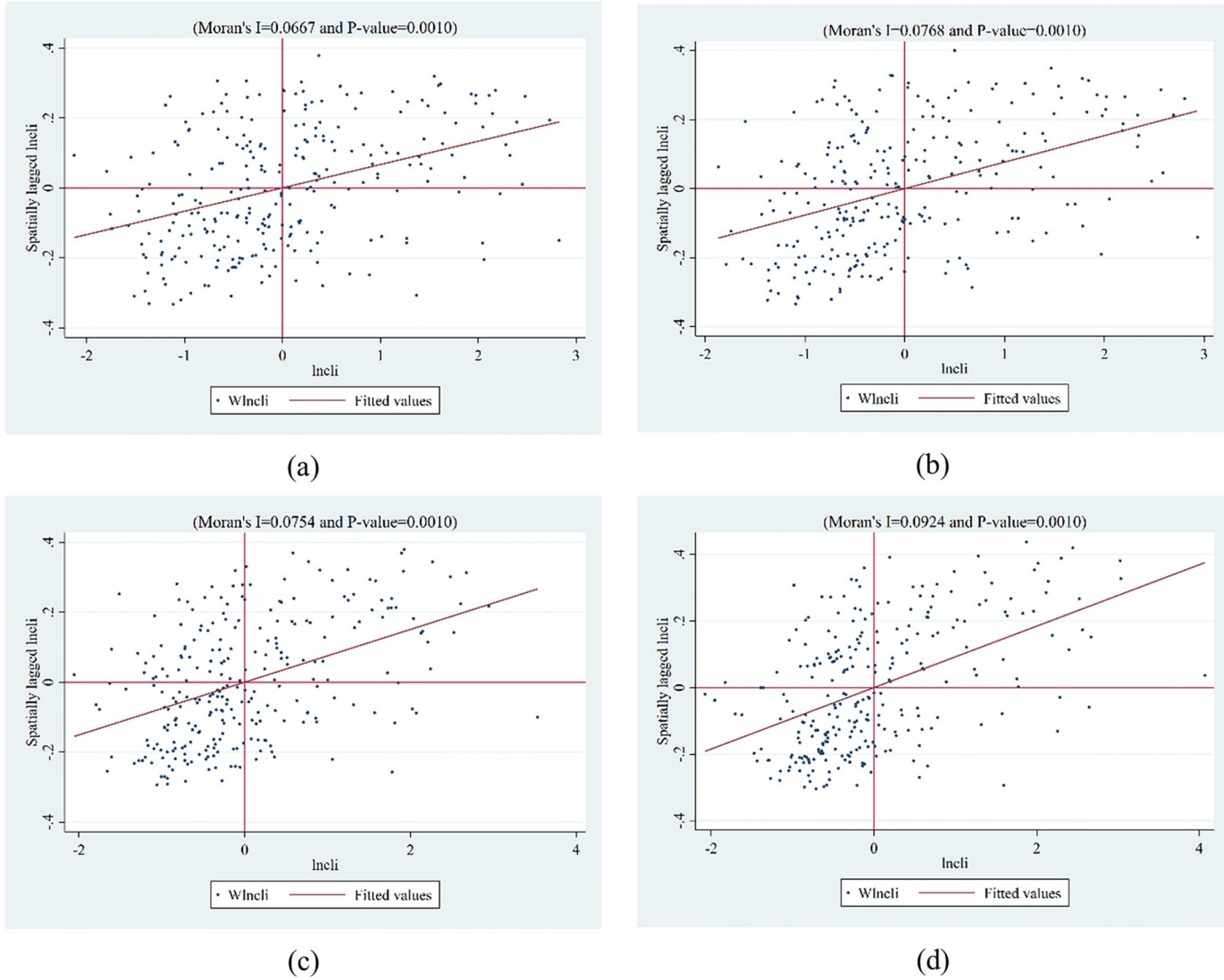

**Fig 4. LISA maps of the CLI in China for 2006, 2010, 2015 and 2019.**

more prominent, i.e., BCP has a better inhibitory effect on CLI in neighboring cities. In regard to the total effect, the inhibitory effect of BCP on CLI is also significant. In addition, the total effect of fixed asset investment on CLI shows inhibition, while the total effect of human capital and the openness on CLI shows exacerbation. Therefore, hypothesis 3 is verified.

Green and low-carbon production technologies, environmental protection concepts and advanced systems can flow to neighboring cities through digital infrastructure [5], thus affecting CLI. First, digital infrastructure builds a sharing network and a platform for cooperation, and with the support of big data, cloud computing and other digital technologies, it can have a "spillover effect" on neighboring regions [81], thus driving the development of neighboring regions and realizing green transformation. Secondly, digital infrastructure accelerates the dissemination of information, broadens people's access to information, raises their awareness of energy saving and environmental protection, reduces carbon emissions at the consumption end, and adjusts the consumption structure [82,90]. Finally, digital infrastructure lays the

**Table 5. Benchmark regression results between BCP and CLI.**

| Variables | (1) | (2) | (3) | (4) | (5) |
|---|---|---|---|---|---|
| | SEM-FE | SLM-RE | SLM-FE | SDM-RE | SDM-FE |
| did | -0.0208** | -0.0233*** | -0.0223*** | -0.0198** | -0.0254*** |
| | (0.00852) | (0.00793) | (0.00853) | (0.00879) | (0.00848) |
| lnpgdp | -0.0375** | -0.0439*** | -0.0347** | -0.0176 | -0.0430*** |
| | (0.0158) | (0.0136) | (0.0151) | (0.0162) | (0.0166) |
| lnfdi | 0.00776*** | 0.00774*** | 0.00696** | 0.00797** | 0.00893*** |
| | (0.00299) | (0.00287) | (0.00290) | (0.00315) | (0.00306) |
| lnstu | 0.0412*** | 0.0356*** | 0.0446*** | 0.0327*** | 0.0396*** |
| | (0.00695) | (0.00663) | (0.00689) | (0.00672) | (0.00690) |
| lnfin | 0.0324** | 0.0111 | 0.0289** | 0.0395*** | 0.0349** |
| | (0.0132) | (0.0100) | (0.0128) | (0.0129) | (0.0137) |
| lnope | 0.0144** | 0.00971* | 0.0155*** | 0.0105* | 0.0151** |
| | (0.00607) | (0.00546) | (0.00601) | (0.00578) | (0.00604) |
| lambda | 0.844*** | | | | |
| | (0.0394) | | | | |
| rho | | 0.955*** | 0.846*** | 0.0168*** | 0.775*** |
| | | (0.00931) | (0.0387) | (0.000407) | (0.0546) |
| Constant | | -0.380*** | | -0.466 | |
| | | (0.0937) | | (0.564) | |

Note: Robust standard errors in parentheses.

*** p<0.01,

** p<0.05,

* p<0.1.

**Table 6. Results of the direct and indirect effects of BCP and CLI.**

| Variables | (1) | (2) | (3) |
|---|---|---|---|
| | LR_Direct | LR_Indirect | LR_Total |
| did | -0.0319*** | -1.799** | -1.831** |
| | (0.00955) | (0.851) | (0.855) |
| lnpgdp | -0.0437*** | -0.0173 | -0.0610 |
| | (0.0155) | (0.580) | (0.579) |
| lnfdi | 0.00837*** | -0.235* | -0.227* |
| | (0.00289) | (0.132) | (0.133) |
| lnstu | 0.0499*** | 2.723*** | 2.773*** |
| | (0.00725) | (0.768) | (0.771) |
| lnfin | 0.0313** | -0.977* | -0.946 |
| | (0.0127) | (0.586) | (0.587) |
| lnope | 0.0224*** | 1.831*** | 1.853*** |
| | (0.00667) | (0.615) | (0.618) |

Note: Robust standard errors in parentheses.

*** p<0.01,

** p<0.05,

* p<0.1.

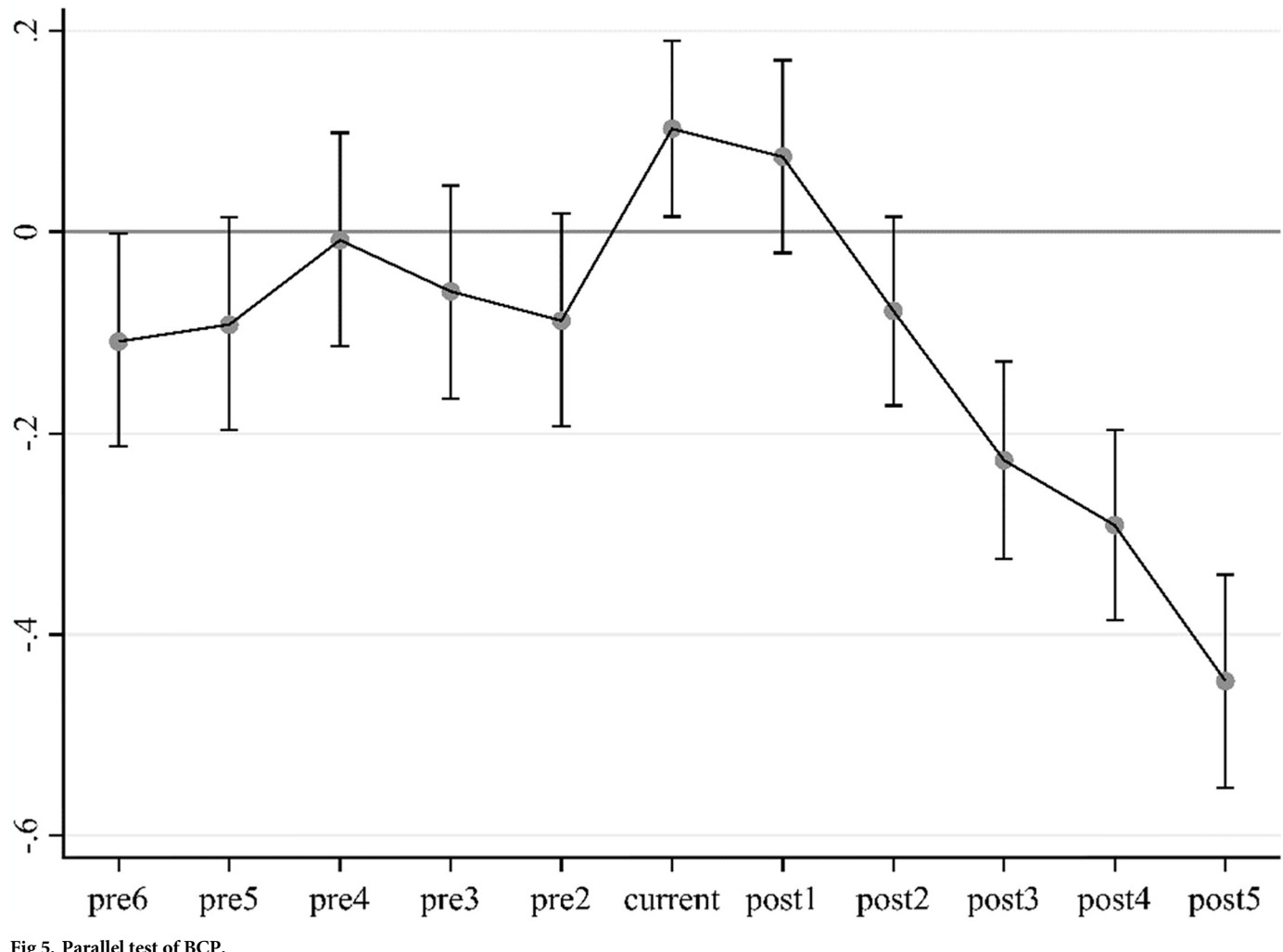

**Fig 5. Parallel test of BCP.**

foundation for the realization of "digital government" and accelerates the transition to new mechanisms, platforms and channels [4]. Currently, digitalization is being seen as an important tool for governance transformation at all levels government [91].

### 4.4 Robust test

**4.3.1 Parallel test.** A valid assumption for the DID method is that it must pass the parallel trend test, i.e., cities in the control group and experimental group have similar trends before the BCP implementation, and the two groups are significantly different after the BCP implementation. So, we first examined the parallel trend test in our robustness test, and the results are shown in Fig 5. Specifically, current in the horizontal coordinate indicates the year of BCP implementation. Since BCP was rolled out in batches, the current is different for different cities, with the current being 2014 for some cities and 2015 or 2016 for others. Since BCP was implemented from 2014, we take 2009 as the start year and 2019 as the end year, so the actual years included in Fig 5 are from 2009 to 2019. From Fig 5, it can be observed that before the BCP implementation, none of the research cities' CLI differ significantly from one another. However, starting from the fourth year (2017), the confidence interval of the CLI coefficient

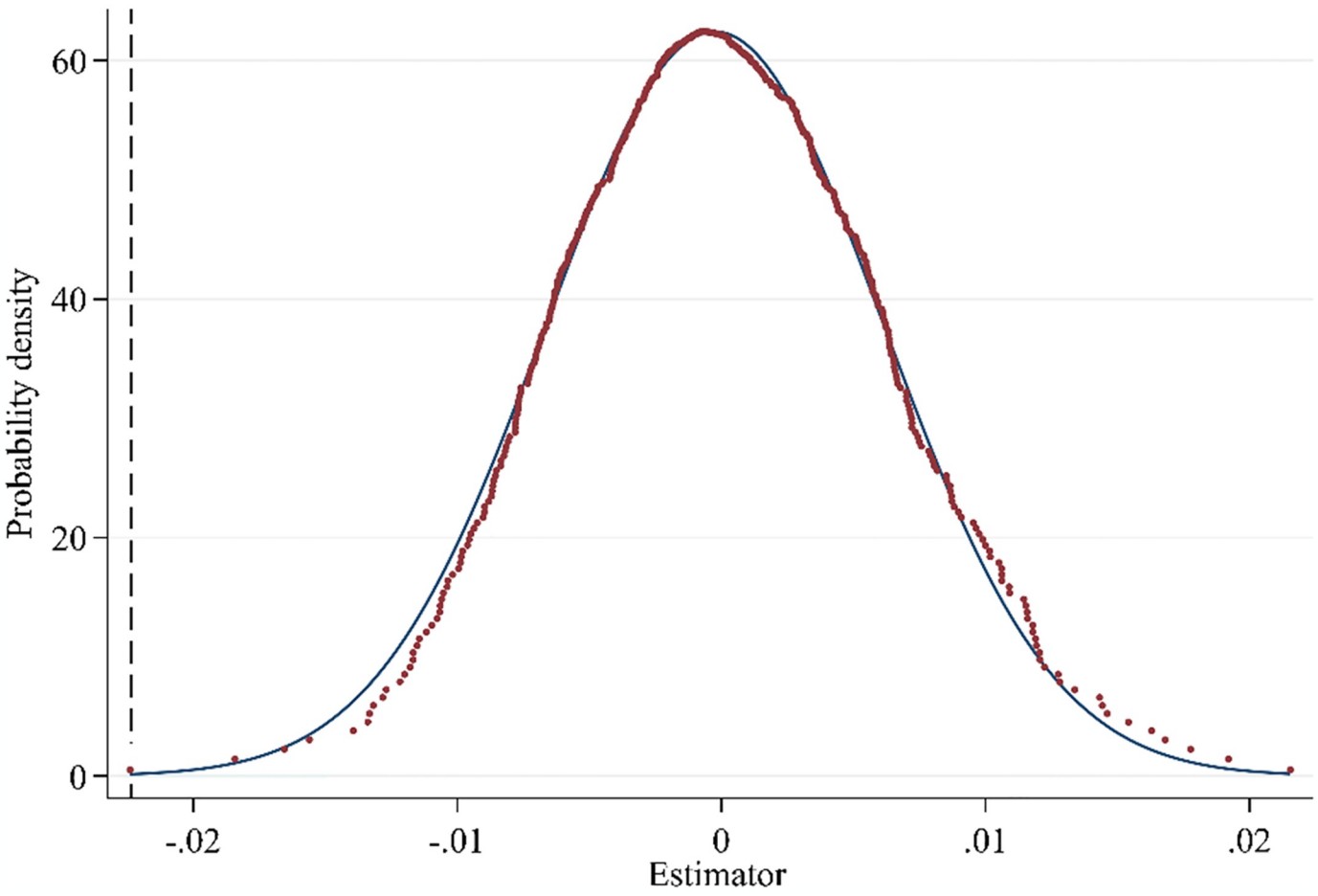

**Fig 6. Distributions of the t value of the estimated coefficients for the placebo test.**

no longer contains 0, demonstrating that following the policy's adoption, the CLI of the pilot cities differs significantly from the non-pilot cities. The implementation effect of BCP has a lag. After the implementation of the policy, it does not produce a restraining effect on CLI immediately but has a significant impact from the fourth year. Therefore, the benchmark regression model is valid, and BCP has a major and delayed inhibitory impact on CLI.

**4.3.2 Placebo test.**  The placebo test is able to test whether changes in CLI are affected by other unknown factors. Therefore, we create new control and experimental groups by randomly choosing cities. Specifically, out of all the sample cities, we choose a random number of BCP pilot cities to create a new experimental group. The remaining cities serve as the new control group. Second, we perform regression estimation based on the new sample and obtain 500 regression coefficients and p-values after repeating the above process 500 times. The kernel density estimates of these 500 regression coefficients are presented in Fig 6. Most of the estimated coefficients are centered around 0, and show a normal distribution, demonstrating that they do not deviate substantially from 0. However, considering that the benchmark regression's real value (-0.0254) is statistically significantly different from this difference, the placebo test appears to be satisfactory. So, the results of the benchmark regression are robust.

**4.3.3 DID method.**  We also conduct robustness tests using the multi-period DID approach, a model that has been widely used in policy evaluation, although it does not take

**Table 7. Robustness test using DID method.**

| Variables | (1) | (2) | (3) | (4) | (5) | (6) |
|---|---|---|---|---|---|---|
| | | | CLI | | | |
| did | -0.257*** | -0.123*** | -0.113*** | -0.113*** | -0.114*** | -0.112*** |
| | (0.0177) | (0.0195) | (0.0197) | (0.0197) | (0.0191) | (0.0193) |
| lnpgdp | | -0.221*** | -0.254*** | -0.261*** | -0.269*** | -0.268*** |
| | | (0.0141) | (0.0167) | (0.0213) | (0.0703) | (0.0702) |
| lnfdi | | | 0.0391*** | 0.0390*** | 0.0387*** | 0.0392*** |
| | | | (0.00677) | (0.00672) | (0.00653) | (0.00650) |
| lnstu | | | | 0.0137 | 0.0132 | 0.0129 |
| | | | | (0.0204) | (0.0194) | (0.0195) |
| lnfin | | | | | 0.00693 | 0.00856 |
| | | | | | (0.0467) | (0.0466) |
| lnope | | | | | | -0.0248 |
| | | | | | | (0.0255) |
| Constant | -2.059*** | 0.224 | 0.192 | 0.115 | 0.116 | 0.426 |
| | (0.00252) | (0.147) | (0.150) | (0.172) | (0.173) | (0.357) |

Note: Robust standard errors in parentheses.

*** $p < 0.01$,

** $p < 0.05$,

* $p < 0.1$.

into account spatial characteristics between variables. Specifically, we conduct a stepwise regression using a fixed-effects model with CLI as the dependent variable and BCP as the independent variable, and the results are shown in Table 7. The estimated coefficient of BCP is -0.112 and significant at the 1% level, suggesting that the CLI in the pilot cities is able to decrease by 11.2% compared to non-pilot cities. The finding is similar to the benchmark regression, i.e., the BCP pilot is able to break the CLI. However, we find a difference in the coefficients between them, which suggests that not taking into account the spatial relationship may bias the assessed policy impacts. Therefore, the SDID model's outputs are more reliable and demonstrate the robustness of the benchmark regression.

**4.3.4 PSM-DID.** This research further utilizes the propensity score matching (PSM) approach for robustness tests to mitigate the estimate bias brought on by the DID and SDID. The steps are as follows: first, a logit regression method is applied to obtain the propensity score value for each city. Second, the city in the treatment group that is most similar to this value is selected as the city where the policy is implemented and regressed for estimation. This approach minimizes systematic differences across cities and thus reduces sample selection bias. Fig 7 depicts the before and after matching, where the propensity score matching and probability densities of the matched treatment and control groups are better than those before matching, indicating better matching. The regression estimates after matching are shown in Table 8. The results in column (1) indicate that the inhibitory effect of BCP on CLI is significant after the PSM-DID method is adopted and remains valid after adding the control variables. Therefore, the benchmark regression results are robust.

**4.3.5 Substitution of spatial weight matrix.** In order to evaluate the robustness of the benchmark regression findings, we replaced the spatial weight matrix. Specifically, we employed the geographic distance weight matrix for model estimation, and the results are shown in Table 9. We find a significant negative relationship between BCP and CLI regardless

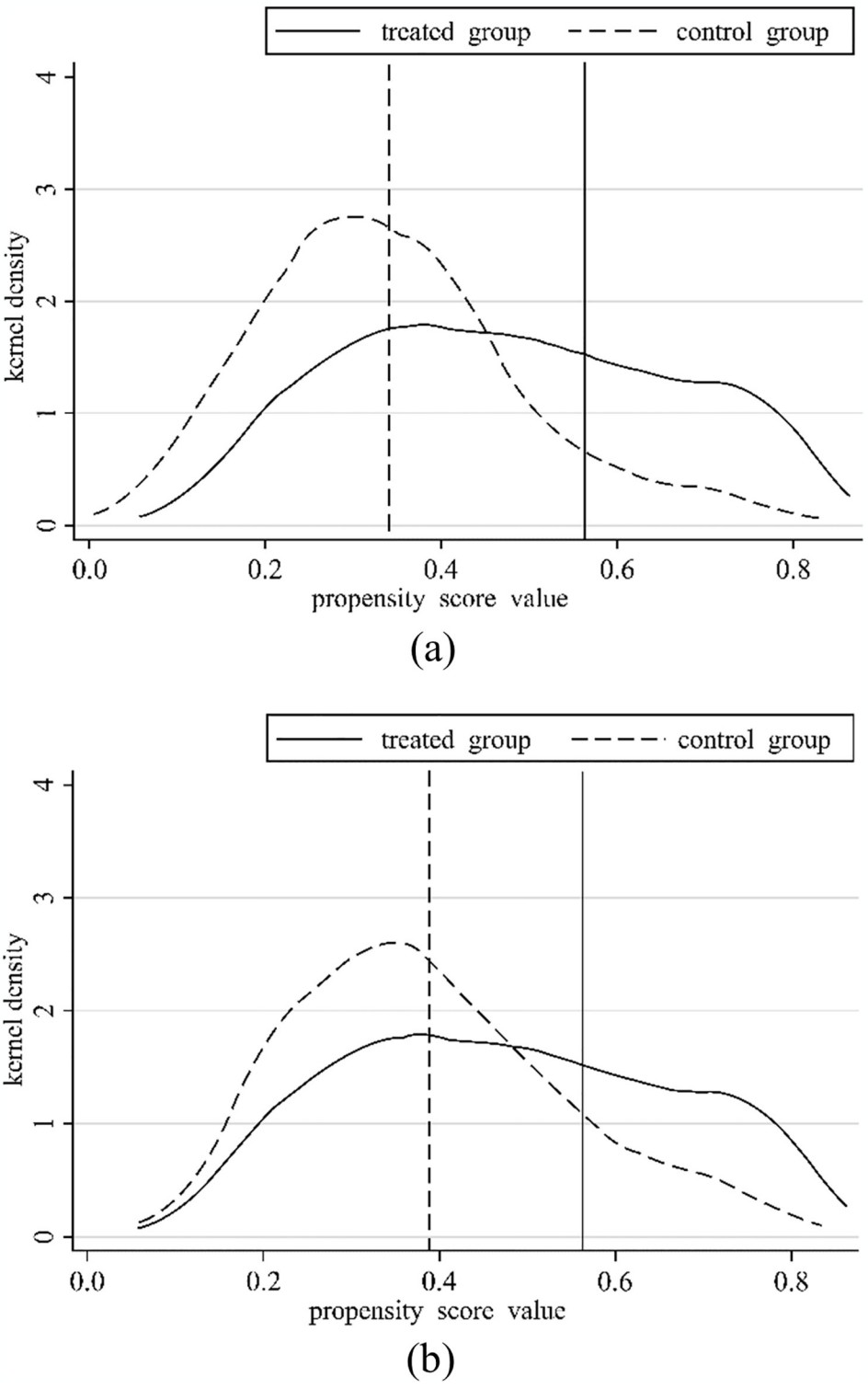

**Fig 7. Before (a) and after (b) matching the probability density function of the propensity score value.**

**Table 8. Robustness test using PSM-DID method.**

| Variables | (1) | (2) |
|---|---|---|
| | **CLI** | |
| did | -0.258*** | -0.109*** |
| | (0.0124) | (0.0122) |
| lnpgdp | | -0.337*** |
| | | (0.0251) |
| lnfdi | | 0.0382*** |
| | | (0.00446) |
| lnstu | | -0.00439 |
| | | (0.0123) |
| lnfin | | 0.0581*** |
| | | (0.0178) |
| lnope | | -0.0219** |
| | | (0.00943) |
| Constant | -2.059*** | 0.626*** |
| | (0.00406) | (0.161) |

Note: Robust standard errors in parentheses.

*** p<0.01,

** p<0.05,

* p<0.1.

**Table 9. Robustness test using other spatial weight matrix.**

| Variables | (1) | (4) | (7) | (10) | (14) |
|---|---|---|---|---|---|
| | **SEM-FE** | **SLM-RE** | **SLM-FE** | **SDM-RE** | **SDM-FE** |
| did | -0.0211** | -0.0295*** | -0.0215** | -0.0206** | -0.0212** |
| | (0.00854) | (0.00841) | (0.00839) | (0.00913) | (0.00855) |
| lnpgdp | -0.0615*** | -0.0358** | -0.0580*** | -0.0150 | -0.0661*** |
| | (0.0156) | (0.0146) | (0.0149) | (0.0156) | (0.0160) |
| lnfdi | 0.00745** | 0.00862*** | 0.00664** | 0.00698** | 0.00633** |
| | (0.00291) | (0.00305) | (0.00285) | (0.00313) | (0.00300) |
| lnstu | 0.0340*** | 0.0462*** | 0.0374*** | 0.0416*** | 0.0302*** |
| | (0.00704) | (0.00705) | (0.00683) | (0.00699) | (0.00702) |
| lnfin | 0.0280** | 0.00980 | 0.0252** | 0.0298** | 0.0230* |
| | (0.0130) | (0.0106) | (0.0126) | (0.0128) | (0.0133) |
| lnope | 0.0177*** | 0.00838 | 0.0149** | 0.00751 | 0.0113* |
| | (0.00603) | (0.00584) | (0.00591) | (0.00607) | (0.00615) |
| lambda | -1.927*** | | | | |
| | (0.126) | | | | |
| rho | | 0.929*** | -2.011*** | 0.832*** | -1.185*** |
| | | (0.0120) | (0.131) | (0.0220) | (0.245) |
| Constant | | -0.581*** | | 4.668*** | |
| | | (0.101) | | (0.970) | |

Note: Robust standard errors in parentheses.

*** p<0.01,

** p<0.05,

* p<0.1.

of which spatial model is applied and whether fixed or random effects are adopted. This suggests that BCP implementation can effectively reduce CLI, and the estimation of the benchmark regression model is robust.

## 4.4 Mediating effect analysis

After benchmark model analysis and robustness test, we find that CLI is significantly negatively impacted by BCP, but how does BCP come to affect CLI? In order to explore this question, this paper adopts the mediation effect model and analyzes the potential impact mechanisms, i.e., technological innovation and industrial upgrading, respectively. Tables 10 and 11 provide summaries of the corresponding outcomes.

Table 9's column (1) demonstrates that BCP significantly affects CLI, column (2) shows that BCP can make the pilot city's green innovation level increase by 33.1%, and column (3) shows that BCP breaks the CLI through technological innovation. Specifically, for every 1% increase in technological innovation, the CLI can be reduced by 3.09%. BCP implementation provides technological support for the improvement of R&D level, promotes the development of digital technology as well as other high-end technological chains and industries, and creates talent aggregation effect and scale effect [12,36,38,92]. Therefore, with the support of digital technology, not only the reduction of energy consumption has been realized [61,74], but also the dependence on traditional resources has been broken [36], which makes the country gradually transition to sustainable energy use, thus breaking the CLI.

Table 11 summarizes the results of the mediation effect analysis using industrial structure upgrading as the mediating variable. Column (1) shows that BCP has a significant negative effect on CLI, column (2) shows that BCP can increase the industrial structure upgrading of

**Table 10. Mediating effect analysis by technological innovation.**

| Variables | (1) | (2) | (3) |
|---|---|---|---|
| | CLI | TEC | CLI |
| did | -0.112*** | 0.331*** | -0.101*** |
| | (0.0193) | (0.0538) | (0.0197) |
| lntec | | | -0.0309*** |
| | | | (0.00960) |
| lnpgdp | -0.268*** | 0.677*** | -0.247*** |
| | (0.0702) | (0.178) | (0.0682) |
| lnfdi | 0.0392*** | -0.0607*** | 0.0374*** |
| | (0.00650) | (0.0193) | (0.00641) |
| lnstu | 0.0129 | 0.129** | 0.0169 |
| | (0.0195) | (0.0531) | (0.0187) |
| lnfin | 0.00856 | 0.953*** | 0.0376 |
| | (0.0466) | (0.123) | (0.0428) |
| lnope | -0.0248 | 0.189*** | -0.0188 |
| | (0.0255) | (0.0530) | (0.0245) |
| Constant | 0.426 | -19.16*** | -0.159 |
| | (0.357) | (0.909) | (0.390) |

Note: Robust standard errors in parentheses.

*** $p < 0.01$,

** $p < 0.05$,

* $p < 0.1$.

**Table 11. Mediating effect analysis by industrial structure.**

| Variables | (1) | (2) | (3) |
|---|---|---|---|
| | CLI | INS | CLI |
| did | -0.112*** | 0.142*** | -0.0398* |
| | (0.0193) | (0.0157) | (0.0226) |
| lnins | | | -0.507*** |
| | | | (0.0653) |
| lnpgdp | -0.268*** | 0.0415 | -0.247*** |
| | (0.0702) | (0.0350) | (0.0633) |
| lnfdi | 0.0392*** | -0.0228*** | 0.0276*** |
| | (0.00650) | (0.00529) | (0.00648) |
| lnstu | 0.0129 | 0.0595*** | 0.0431** |
| | (0.0195) | (0.0168) | (0.0178) |
| lnfin | 0.00856 | 0.0510** | 0.0344 |
| | (0.0466) | (0.0248) | (0.0414) |
| lnope | -0.0248 | 0.00999 | -0.0198 |
| | (0.0255) | (0.0248) | (0.0165) |
| Constant | 0.426 | 1.952*** | 1.415*** |
| | (0.357) | (0.339) | (0.296) |

Note: Robust standard errors in parentheses.

*** $p<0.01$,

** $p<0.05$,

* $p<0.1$.

pilot cities by 14.2%, and column (3) shows that BCP can break the CLI through industrial structure upgrading. Specifically, for every 1% increase in industrial structure upgrading, the CLI can be reduced by 50.7%. BCP implementation on the one hand brings new business forms and modes, effectively plays the role of data as a factor of production, drives industrial upgrading and optimizes the industrial structure [90]. On the other hand, it can increase the carbon intensity through optimizing the division of labor and technological reform, etc., and promote the low-carbon transformation [93]. BCP makes the proportion of the tertiary industry continue to increase and can reduce emissions and environmental protection while upgrading and innovating existing products and technologies, thus breaking the CLI [35,39,87]. Therefore, Hypothesis 2 is verified.

## 5. Heterogenous analysis

### 5.1 Heterogenous analysis of resource endowments

According to the National Sustainable Development Plan for Resource Based Cities (2013–2020), we classify all the sample cities into two categories, i.e., resource cities and non-resource cities, and estimate the policy effects separately, and the outcomes are displayed in Table 12. Via column (1) of Table 11, it can be seen that BCP can lower the CLI of non-resource cities by 11.2%, according to the coefficient of did, which is -0.112 and significant at the 1% level, whereas this coefficient is 12.3% for resource cities and is significant at the 1% level. This shows that resource-based cities see a greater policy impact of BCP on CLI. The potential cause of this is that resource cities are prone to face the "resource curse" in the long-term development, and therefore they are more eager to look for ways of low-carbon transition and

Table 12. Heterogenous analysis of resource endowments.

| Variables | (1) | (2) |
|---|---|---|
| | Non-resources city | Resources city |
| did | -0.112*** | -0.123*** |
| | (0.0244) | (0.0310) |
| lnpgdp | -0.326*** | -0.193** |
| | (0.0494) | (0.0964) |
| lnfdi | 0.0518*** | 0.0330*** |
| | (0.00954) | (0.00862) |
| lnstu | -0.00201 | 0.0233 |
| | (0.0292) | (0.0256) |
| lnfin | 0.0713** | -0.0744 |
| | (0.0323) | (0.0664) |
| lnope | -0.0662* | -0.00575 |
| | (0.0342) | (0.0280) |
| Constant | 0.691 | 0.627 |
| | (0.502) | (0.421) |
| Obs. | 2240 | 1484 |

Note: Robust standard errors in parentheses.

*** p<0.01,

** p<0.05,

* p<0.1.

green development. In this situation, resource cities may use digital technology to their greatest advantage to increase the efficiency of resource usage, encourage energy conservation and emission reduction, and lessen the tension between economic growth and environmental preservation. Thus, resource-based cities see a greater influence of BCP on CLI, which also highlights the latecomer advantage of future transformation and development of resource-based cities.

## 5.2 Heterogenous analysis of city location

Geographical disparities in China's enormous nation have an impact on the degree of economic growth among cities. Therefore, according to the classification of each province by the National Bureau of Statistics and the region to which it belongs, we divide the sample cities into two categories based on their geographic locations, i.e., cities in the central and western regions and cities in the eastern regions, then estimate the policy effects separately, and the results are shown in Table 13. From column (1) we find that in the eastern region, BCP may lower the CLI of the pilot cities by 12.54% and is significant at the 1% level, and this coefficient is 11.17% in the pilot cities in the central and western regions, which indicates that the policy effect of BCP is more significant in the cities in the eastern. The reason for this is that the eastern region is economically developed, with better infrastructure and a higher rate of application and penetration of digital technology. This gives the eastern region a greater advantage in the construction and application of broadband networks and enables it to more effectively utilize digital technologies to enhance energy efficiency and resource use, and to achieve both productivity and carbon efficiency. In addition, it is easier for the eastern region to obtain policy support and financial investment to promote the green transformation of digital infrastructure. For example, by investing in distributed and centralized renewable energy projects,

**Table 13. Heterogenous analysis of city location.**

| Variables | (1) | (2) |
|---|---|---|
| | **Eastern cities** | **Central and western cities** |
| did | -0.125*** | -0.111*** |
| | (0.0384) | (0.0240) |
| lnpgdp | -0.268*** | -0.271*** |
| | (0.0515) | (0.0990) |
| lnfdi | 0.00871 | 0.0299*** |
| | (0.0123) | (0.0100) |
| lnstu | -0.0275 | 0.00697 |
| | (0.0355) | (0.0212) |
| lnfin | 0.00912 | 0.0663 |
| | (0.0378) | (0.0659) |
| lnope | -0.244*** | -0.0398 |
| | (0.0424) | (0.0277) |
| Constant | 4.580*** | 0.0423 |
| | (0.657) | (0.361) |
| Obs. | 1750 | 1974 |

Note: Robust standard errors in parentheses.

*** p<0.01,

** p<0.05,

* p<0.1.

purchasing renewable energy on a market basis, and subscribing to green power certificates, the eastern region can more effectively reduce carbon emissions from digital infrastructure.

## 5.3 Heterogenous analysis of digital base

The number of urban Internet broadband access users can reflect the coverage and access of a region's information infrastructure, which is one of the important foundations for the development of digital economy. Therefore, according to the median of this indicator, we divided the sample cities into two categories, one is the city with relatively perfect information infrastructure, the other is the city with poor information infrastructure, and estimated the policy effect respectively, as shown in Table 14. The coefficient of did in column (1) is -0.5543 and is significant at 1% level, however, this coefficient is not significant in column (2), which shows that BCP has a considerable impact on CLI in places with superior information infrastructure, while the policy effect of BCP is not significant in cities with poor information infrastructure. This finding is obvious that having good information infrastructure makes it easier to build a digital system driven by data as innovation and based on communication networks, which will play a great role in urban innovation, digital operation, and industrial transformation.

## 6. Conclusions and discussion

### 6.1 Discussion

Our key findings suggest that BCP make a significant contribution to breaking the CLI, which is consistent with existing literature supporting digital infrastructure as a way to reduce carbon emissions. Digital infrastructure drives green technology innovation, improves resource efficiency while reducing environmental pollution. More importantly, it brings about a change in

**Table 14. Heterogenous analysis of the city's digital base.**

| Variables | (1) | (2) |
|---|---|---|
| | **Better digital base** | **Poorer digital base** |
| did | -0.0554** | -0.0434 |
| | (0.0253) | (0.0321) |
| lnpgdp | -0.506*** | -0.125* |
| | (0.0592) | (0.0709) |
| lnfdi | 0.0453*** | 0.0199** |
| | (0.00979) | (0.00792) |
| lnstu | -0.0505 | 0.0263 |
| | (0.0512) | (0.0232) |
| lnfin | 0.0627 | -0.0266 |
| | (0.0427) | (0.0463) |
| lnope | -0.0797* | 0.0445* |
| | (0.0449) | (0.0240) |
| Constant | 3.705*** | -1.453*** |
| | (0.690) | (0.376) |
| Obs. | 1,862 | 1,862 |

Note: Robust standard errors in parentheses.

*** $p<0.01$,

** $p<0.05$,

* $p<0.1$.

the way the city develops, altering its development model that is overly reliant on traditional resources, thus breaking the CLI [36,39,43,45]. When the traditional development model continues, the output performance of high pollution, high emission and low efficiency will further aggravate CLI and even affect climate change [21,22]. Therefore, reducing CLI should shift from an economic model with high energy consumption and high carbon emissions to a low-carbon circular development model with high efficiency and high output. Fortunately, this shift can be realized through the digital development strategy.

At the same time, BCP accelerates technological innovation and industrial structure upgrading, thus contributing to carbon unlocking. This research adds well to existing literature. In the extensive discussion of the influential factors of CLI, the positive effects of digitalization have been overlooked in the literature [27,33,36–39]. These findings provide new perspectives by revealing how digital infrastructure can break the CLI through technological innovation and industrial structure upgrading. The advancement of digital technology has been an important catalyst for technological innovation, industrial transformation and consumption upgrading [3,12,46,82].

More importantly, on the one hand, our findings verify the importance of digital infrastructure in the coordinated development of regional economies, that is, it is crucial in promoting the coordination of inter-regional environmental governance; on the other hand, the policy lag effect found in this study provides an important time reference dimension for policy-makers, and also emphasizes the need to consider medium- and long-term impacts when evaluating the implementation effect of such programs. However, the above two key issues have never been discussed in previous studies.

Moreover, the heterogeneity effect of digital infrastructure provides theoretical support for the development of regionally differentiated policies, i.e., the need to take into account local

characteristics prior to policy implementation. In summary, this study supplements the research gap between digital infrastructure and CLI, providing empirical support and scientific basis for policy development. These findings, on the one hand, provide city managers with a more comprehensive and in-depth perspective to help formulate scientific and effective low-carbon development policies, and on the other hand, as the digital transformation deepens, its potential to promote environmentally sustainable development will be further explored and utilized.

## 6.2 Conclusion

Based on an empirical analysis of 266 prefecture-level cities in China from 2006 to 2019, this study explores the effects, mechanisms, synergies and heterogeneity of the BCP on CLI. The results are as follows:

1. BCP can significantly reduce CLI, and after a number of robust tests, the finding is still valid.

2. The impact of BCP on CLI has a spatial spillover effect, that is, BCP can contribute to CLI in both neighboring and local cities.

3. There is a lag in the policy effect of BCP, i.e., its inhibitory effect on CLI becomes significant only from the third year of policy execution.

4. The contribution of BCP to CLI is more significant in the eastern region, resource-based cities, and cities with a better digital base.

5. BCP is able to break CLI by promoting green technological innovation and facilitating industrial structure upgrading. With the increasing severity of the global climate change problem and the pursuit of sustainable development, achieving carbon neutrality has become a common challenge faced by various countries. As a policy initiative covering economic, social and environmental dimensions, the impact of BCP on carbon emissions is not only related to the realization of China's future carbon emission reduction targets, but also provides empirical support and policy reference for global carbon management and climate change governance. By systematically studying the carbon locking effect of BCP, we can not only provide theoretical support and practical guidance for the formulation and implementation of similar policies worldwide, but also further address global carbon management, so as to solve the environmental challenges brought by climate change and promote the development and realization of sustainable development goals in other countries and regions.

## 7. Policy recommendations

In light of the foregoing results, we offer the following policy recommendations. First, the central government should actively promote the construction of digital infrastructure construction, enhance the breadth and depth of regional informatization. Accelerate the penetration of digitalization into traditional industries, strengthen its wide application in various fields such as production and life, and maximize the sharing of digital dividends.

Second, local governments should actively encourage inter-regional exchanges and cooperation in digital technology and low-carbon green transformation, remove obstacles to the flow of industrial inputs between regions. Optimize the allocation of resources between regions to create regional aggregates with higher levels of digital technology development and higher

green economy efficiency. Make full use of the spatial spillover effect of digitalization to create a good demonstration effect among regions.

Third, when planning and designing digital infrastructure, government departments should provide clearer support services for emerging industrial enterprises and accelerate the agglomeration of emerging enterprises, so as to realize the optimization and upgrading of industrial structure and technological innovation. In addition, the knowledge diffusion effect of digital infrastructure can be fully realized through the construction of innovative service platforms and other means.

Fourth, take the policy as an opportunity to increase the government's investment in the science and technology innovation sector. On the one hand, innovative talents should be cultivated, and on the other hand, intelligent innovation platforms should be established to enhance the positive externalities of scientific and technological innovation capacity in environmental governance. At the same time, actively advocate and strengthen the concept of green production and green consumption, the implementation of strict environmental regulatory policies, forcing enterprises to green low-carbon upgrading, which in turn promotes the energy consumption transformation in the low-carbon direction, and ultimately break the CLI. Fifth, resource-based cities should make use of digital technological innovation to improve resource efficiency, thereby promoting energy conservation and emission reduction; non-resource-based cities should pay attention to digital infrastructure and promote the optimization and upgrading of industrial structure. Eastern regions should further leverage their digital foundation advantages to promote the research, development and application of low carbon technologies, while central and western regions should further improve their digital infrastructures to enhance the quality and efficiency of the regional economy. For cities with better digital infrastructure, governments should focus on promoting technological innovation and achieving efficient use of energy and resources; for cities with poorer digital infrastructure, governments need to invest more in upgrading infrastructure and encouraging enterprises and individuals to adopt digital means of production and life.

Despite some findings, this study still has some shortcomings. Due to the availability of some indicators, this paper has not been able to conduct a more detailed indicator system to measure CLI. Future research needs to be carried out in the following aspects: First, a more refined carbon lock-in index system needs to be developed. Combined with regional and industry characteristics, the extent of carbon lock-in effect and its influencing factors should be assessed. Secondly, the case studies of different regions are integrated to deeply discuss the implementation effect of BCP in different cities and analyze the feasibility and necessity of policy adjustment. Furthermore, a comprehensive framework between digital economy and sustainable development should be constructed, the paths of synergistic development of the two mentioned above should be explored, and targeted policy recommendations should be put forward. Finally, a cross-country comparison is conducted to analyze the differentiated impact of digital development on carbon lock-in in different national contexts.

## Author Contributions

**Conceptualization:** Liang Guo.

**Data curation:** Lijing Chen.

**Formal analysis:** Lijing Chen.

**Investigation:** Lijing Chen.

**Methodology:** Zhen Yang.

**Resources:** Liang Guo.

**Software:** Zhen Yang.

**Supervision:** Liang Guo.

**Validation:** Zhen Yang.

**Visualization:** Lijing Chen.

**Writing – original draft:** Zhen Yang.

**Writing – review & editing:** Liang Guo.

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
