## [Decision Letter · Decision Letter 0]

14 Jun 2024

PONE-D-24-09583The urban carbon unlocking effect of digital infrastructure construction: A spatial difference-in-difference analysis from ChinaPLOS ONE

Dear Dr. Chen,

Thank you for submitting your manuscript to PLOS ONE. After careful consideration, we feel that it has merit but does not fully meet PLOS ONE’s publication criteria as it currently stands. Therefore, we invite you to submit a revised version of the manuscript that addresses the points raised during the review process.

We look forward to receiving your revised manuscript.

Kind regards,

Bijay Halder

Academic Editor

PLOS ONE

Journal Requirements:

3. We note that Figure 3 in your submission contain [map/satellite] images which may be copyrighted. All PLOS content is published under the Creative Commons Attribution License (CC BY 4.0), which means that the manuscript, images, and Supporting Information files will be freely available online, and any third party is permitted to access, download, copy, distribute, and use these materials in any way, even commercially, with proper attribution. For these reasons, we cannot publish previously copyrighted maps or satellite images created using proprietary data, such as Google software (Google Maps, Street View, and Earth). For more information, see our copyright guidelines: http://journals.plos.org/plosone/s/licenses-and-copyright.

a. You may seek permission from the original copyright holder of Figure 3 to publish the content specifically under the CC BY 4.0 license.  

Reviewers' comments:

Reviewer's Responses to Questions

**Comments to the Author**

1. Is the manuscript technically sound, and do the data support the conclusions?

Reviewer #1: Yes

Reviewer #2: Yes

2. Has the statistical analysis been performed appropriately and rigorously? 

Reviewer #1: Yes

Reviewer #2: Yes

3. Have the authors made all data underlying the findings in their manuscript fully available?

Reviewer #1: Yes

Reviewer #2: Yes

4. Is the manuscript presented in an intelligible fashion and written in standard English?

Reviewer #1: No

Reviewer #2: Yes

5. Review Comments to the Author

Reviewer #1: This manuscript investigated the carbon lock-in phenomenon in the economic development of modern countries and how digital infrastructure construction addresses this issue. Taking the “Broadband China” policy as an example, this manuscript raises an interesting topic, which is of great significance to countries seeking structural transformation of their economies, especially developing countries like China.

However, it appears that the focus of this manuscript may have been misplaced. The manuscript in question examines the impact of BCP on CLI. While this issue is undoubtedly of great importance to China's economic development, such a narrow focus may diminish the academic merit of the study. This study would be more suitable for publication in a Chinese journal unless the authors decide to shift the focus of the study to reveal the impact of digital infrastructure on CLI and explore this phenomenon with China as the study area.

The logical structure of the paper is unclear, making it challenging to follow. For example, the INTRODUCTION section is excessively lengthy, the METHODS section is too brief, and the discussions are mixed in the RESULTS section. This greatly reduces the readability of this article.

Lastly, improvements are needed in the language used throughout the manuscript. Considering these issues, I cannot recommend accepting the current version of the manuscript.

Here are other issues worth considering:

1. The significance of digital infrastructure is unduly emphasized in the Introduction. It is recommended that the article's focus be shifted to the subject matter at hand as soon as possible. The author's comprehensive introduction provides a detailed account of the Chinese government's stance on digital infrastructure and the initiatives undertaken by China. However, this appears to be tangentially related to the subject matter of the study.

It is recommended that the introduction section include more detailed information on the relationship between carbon emissions, digital infrastructure, and economic development. The significance of focusing on the unlocking effect can then be further elucidated by presenting the phenomenon of carbon lock-in.

2. The authors wish to discuss a multitude of issues, which could be more effectively addressed by grouping them into two. For instance, the first and fourth issues are, in fact, both concerned with the impact of digital facility construction on carbon lock-in.

3. In section 2.2 of LITERATURE REVIEW, the majority of the articles referenced appear to originate from China. It is recommended that further studies from other regions, such as Europe and North America, be cited in order to provide a more comprehensive understanding of the subject matter.

4. In the Results section, the authors discuss the spatial spillover effect as one of the topics addressed in their analysis. Nevertheless, this topic is not addressed in the hypothesis section. A discrepancy exists between the content of the hypothesis and the analysis presented in the results section.

5. The Methods section does not present the full range of methods employed in the article and the role of these methods in relation to each other. While it is acceptable to detail only the main methods in this section, it is highly recommended that a figure be included to illustrate the methodology of the research.

6. It is recommended not to mix the results section with the discussion section. The summary in the conclusions section appears somewhat abrupt due to the lack of a detailed discussion of the topic. Furthermore, the policy recommendations lack sufficient support.

Reviewer #2: The manuscript addresses a very meaningful topic. The objectives are stated clearly. However, this manuscript is not yet ready for publication. There are some suggestions that the authors need to address before the manuscript can be considered for publication.

1. Some sentences are lengthy and contain multiple ideas. It's recommended to break them into shorter sentences to enhance readability or consider professional proofreading. For instance, "The aim of the research is to statistically assess the effects of BCP on CLI and its mechanisms." could be simplified to "This research aims to statistically assess the effects and mechanisms of BCP on CLI."

2. The introduction in Chapter 1 and the conclusion section in Chapter 6 lack clear statements regarding the research's contributions, significance, and motivation.

3. In Fig. 1, the mechanism diagram illustrating how BCP influences CLI serves as the basis for the experimental section of the article, but it seems to lack explanations for the content of the figure (steps, groups, arrows, and the meaning and sources of each element). It's advisable to clarify the logic and associations with the four indicators quantifying CLI in Section 3.2, enhancing the article's theoretical logic and reliability.

4. In Section 5.3, the heterogeneous analysis, it's essential to clearly describe the classification criteria first. Explain how the sample cities are categorized, including criteria such as resource endowment, geographical location, and digital infrastructure. This will aid readers in understanding the research's methodology and logic, ensuring the interpretability of the results. Furthermore, elaborating on the practical implications of the heterogeneous results is suggested. While the author mentions the policy effects on different categories of cities in the results section, further explanation of the practical implications of these findings, possibly in the subsequent policy recommendations, is advisable.

5. The conclusion section summarizes the research's purpose, key findings, and policy recommendations. However, it's important to ensure logical coherence and organization. It's suggested to add a discussion section, closely linking each conclusion with corresponding evidence or results, or comparing and echoing existing research, to help readers better understand the research's significant findings.

6. For future prospects, it's recommended to provide anticipated results and implications, guiding and inspiring more scholars to conduct further research.

6. PLOS authors have the option to publish the peer review history of their article (what does this mean?). If published, this will include your full peer review and any attached files.

Reviewer #1: No

Reviewer #2: No

---

## [Author Response · Author response to Decision Letter 0]

29 Jul 2024

Response to Reviewers Comments

Dear editor and reviewers:

On behalf of my co-authors, we thank you very much for your letter and for the reviewers’ comments concerning our manuscript entitled “The urban carbon unlocking effect of digital infrastructure construction: A spatial difference-in-difference analysis from China (PONE-D-24-09583)”. Those comments are all valuable and very helpful for revising and improving our paper, as well as the important guiding significance to our research. We have studied the comments carefully and have made a major revision and re-plan which we hope to meet with the approval. We tried our best to improve the manuscript and all changes made in the revised manuscript are highlighted in blue that will not influence the content and the framework of the paper. The point-by-point responds to reviewers’ comments are as follows:

Reviewer # 1 comment:

1. This manuscript investigated the carbon lock-in phenomenon in the economic development of modern countries and how digital infrastructure construction addresses this issue. Taking the “Broadband China” policy as an example, this manuscript raises an interesting topic, which is of great significance to countries seeking structural transformation of their economies, especially developing countries like China.

However, it appears that the focus of this manuscript may have been misplaced. The manuscript in question examines the impact of BCP on CLI. While this issue is undoubtedly of great importance to China's economic development, such a narrow focus may diminish the academic merit of the study. This study would be more suitable for publication in a Chinese journal unless the authors decide to shift the focus of the study to reveal the impact of digital infrastructure on CLI and explore this phenomenon with China as the study area.

Response 1: We would like to thank you for kindly reminding us. The revisions are as follows here for your convenience.

The research in this paper focuses on the impacts and mechanisms of BCP on city-level CLI. First, the findings provide a theoretical basis and practical examples for understanding the role of digital transformation in promoting low-carbon development. Second, the findings provide a quantitative assessment of policy effects for policymakers, which can help optimize and adjust existing policies for a more effective low-carbon transition. Further, the study emphasizes the impact of regional development differences on policy effectiveness, providing a new perspective for achieving coordinated regional development and balance. Finally, the study emphasizes the key role of technological progress and industrial structure upgrading in digital transformation. Most importantly, the results of the study add to the research on digital infrastructure and low carbon development, providing new ideas for low carbon transformation at the city and national levels, i.e., facilitating the optimization of economic structure and greening of energy consumption through digital technology facilities.

In fact, the Broadband China Strategy is one of China’s digital infrastructure construction projects. BCP’s goal is to build a high-speed, secure and reliable broadband network environment that supports and promotes the digital transformation and informatization development of the entire country through the construction and upgrading of broadband networks. Digital infrastructure construction includes various infrastructure elements, such as broadband networks, data centers, and cloud computing infrastructure. The Broadband China Strategy pays special attention to the construction and popularization of broadband networks among these elements, as broadband networks are an important foundation for the development of the information society and digital economy. Therefore, Broadband China Strategy is an essential part of China’s digital infrastructure construction, which plays a key role in digital transformation and has a far-reaching impact on the digital upgrading of China’s economy and the process of social informatization. In summary, BCP, as one of the keys to digital infrastructure construction, and its investigation of green and low-carbon transformation and development of cities is a good complement and improvement to digital infrastructure research, providing valuable experience and inspiration for digital infrastructure construction and sustainable development worldwide.

Taking China as an example, the study provides new ideas and Chinese experience for other countries in strengthening environmental regulation and promoting sustainable development, and supports the potential of digital infrastructure to promote international cooperation, especially in promoting the development of global low-carbon technologies and green finance, which provides a basis for cooperation in the international community’s joint response to climate change. More importantly, the findings of the study support the idea that digital infrastructure development can achieve a win-win situation for both environmental protection and economic growth, which is of strategic importance for other countries to protect the environment while pursuing economic growth.

2. The logical structure of the paper is unclear, making it challenging to follow. For example, the INTRODUCTION section is excessively lengthy, the METHODS section is too brief, and the discussions are mixed in the RESULTS section. This greatly reduces the readability of this article.

Response 2: Thank you for your thought-provoking suggestion. 

According to your comments, we have made the following modifications. First, we have cut the content of the introduction appropriately. Second, we have adjusted and deepened it to make this part not only more logical, but also closely related to the research topic. Furthermore, we supplement the methods used in this study and present them in the form of a methodological flow. Finally, we rewrote the conclusion to include not only a broad discussion of the findings, but also feasible policy recommendations based on the findings. We believe that the readability of the manuscript can be greatly improved after the above adjustments.

3. Improvements are needed in the language used throughout the manuscript.

Response 3: We would like to thank you for kindly reminding us.

We have polished the manuscript to improve the level of language expression, and we have revised, improved and deepened the content of the manuscript. We believe this will improve the readability of the manuscript.

4. The significance of digital infrastructure is unduly emphasized in the Introduction. It is recommended that the article's focus be shifted to the subject matter at hand as soon as possible. The author's comprehensive introduction provides a detailed account of the Chinese government's stance on digital infrastructure and the initiatives undertaken by China. However, this appears to be tangentially related to the subject matter of the study. It is recommended that the introduction section include more detailed information on the relationship between carbon emissions, digital infrastructure, and economic development. The significance of focusing on the unlocking effect can then be further elucidated by presenting the phenomenon of carbon lock-in.

Response 4: Thank you for your kind suggestion which is of great help to the integrity of our manuscript.

Based on your suggestions, we have revised the introduction. On the basis of fully introducing the importance of digital infrastructure, we summarize its research progress related to carbon emissions, and further propose the significance of the improvement and layout of digital infrastructure to change the traditional development mode and break the carbon lock-in. We believe that the revised introduction will be more logical, thus increasing the readability of the manuscript. The revisions are as follows here for your convenience.

L99-121 (Page 5 in all marked versions):

Many scholars believe that digital infrastructure has the potential to be sustainable and are optimistic about reducing carbon emissions. Romm (2002) found that widespread use of the Internet resulted in a significant reduction in energy intensity in the United States and highlighted the important contribution of ICTs in reducing greenhouse gases. Moyer and Hughes (2012) found that digital infrastructure could reduce global carbon emissions by about 50 years. Liu et al. (2015) verified that digital infrastructure can reduce carbon emissions in China, and there is significant regional heterogeneity in this effect. Asongu et al. (2018) found that digital infrastructure layout has a positive impact on reducing carbon emissions in African countries. Zahra et al. (2019) demonstrate that ICTs can reduce carbon emissions in Iran’s transport sector. Zahra et al. (2019) found that ICT generally reduces carbon emissions in countries along the Belt and Road. On the one hand, digital infrastructure can optimize the industrial structure and promote industrial transformation and upgrading. The reduction of energy use intensity in the service sector helps to reduce carbon emissions. On the other hand, digital infrastructure can promote technological progress, accelerate the research and development and diffusion of innovation, facilitate the development of clean energy, improve energy utilization efficiency, strengthen the monitoring and control of carbon emissions, and thus reduce carbon emissions. It can be seen that digital infrastructure helps to change the traditional development model from relying on large-scale energy consumption to a more low-carbon and sustainable development mode, effectively breaking the carbon lock of traditional industries, and providing key technical and structural support for the global realization of a low-carbon economic path.

5. The authors wish to discuss a multitude of issues, which could be more effectively addressed by grouping them into two. For instance, the first and fourth issues are, in fact, both concerned with the impact of digital facility construction on carbon lock-in.

Response 5: We thank you for this careful comment.

Based on your suggestions, we have integrated the research questions to ensure the readability of the manuscript. The revisions are as follows here for your convenience.

L157-163 (Page 8 in all marked versions):

In this regard, China’s digital transformation and its commitment to the “dual carbon” goal make it a good practical case to study the BCP-CLI relationship. Therefore, we cannot help but ask: (1) How does BCP break the China’s CLI? (2) Does BCP have spatial spillover effects and heterogeneity on CLI? The motivation of this study is to answer the above questions and investigate to what extent BCP can influence CLI.

6. In section 2.2 of LITERATURE REVIEW, the majority of the articles referenced appear to originate from China. It is recommended that further studies from other regions, such as Europe and North America, be cited in order to provide a more comprehensive understanding of the subject matter.

Response 2: We would like to thank you for this insightful comment.

In response to your comments, we have added the research literature on the relationship between digital development and carbon emissions in Europe and North America in the literature review section to provide a more comprehensive understanding of this topic. The revisions are as follows here for your convenience.

L249-272 (Page 12 in all marked versions):

Numerous studies have looked at how digital technologies affect carbon emissions to date (Dong et al. 2022a, Hieu, Le Thanh and Anh 2023, Zhang et al. 2022a, Zhang et al. 2022b), but there is still some controversy. For example, Sadorsky (2012) argues that information and communications technology (ICT) development can raises power usage and carbon emissions. This conclusion is supported by the investigations of Collard, Fève and Portier (2005), who find that ICT do not contribute to improved energy usage and efficiency. Salahuddin and Alam (2016) also fails to identify any benefits of ICT for energy using panel data from OECD countries. Khan (2019), based on data from N-11 countries, found that the use of ICT devices actually increased carbon emissions. However, Lu (2018) found a positive carbon mitigation effect of ICT in 12 Asian countries and considered it as an important strategy to achieve low carbon development. Raheem, Tiwari and Balsalobre-Lorente (2020) demonstrated the G7 countries’ ability to reduce carbon emissions in the long run thanks to ICT. Škare, Gavurova and Porada-Rochon (2024) examines the impact of digitalization on the carbon footprint of governments, households, businesses, NGOs and imports across the EU and draws positive conclusions. Bocean (2023) assesses the positive impact of digital transformation on the economic performance and sustainability of EU countries. Based on data at a global level, Zuo et al. (2024) found that the impact of digital development on carbon reduction is much stronger in Europe and North America than in other countries. In addition, a few scholars believe that the impact of ICT on carbon emissions is not significant. Based on Tunisia’s long time series data, Amri, Zaied and Lahouel (2019) found that ICT has negligible impact on carbon emissions, which means that the Tunisian government can promote the development of ICT.

7. In the Results section, the authors discuss the spatial spillover effect as one of the topics addressed in their analysis. Nevertheless, this topic is not addressed in the hypothesis section. A discrepancy exists between the content of the hypothesis and the analysis presented in the results section.

Response 7: Thank you for your helpful comment. 

We added the hypothesis that BCP has a spatial spillover effect on CLI (hypothesis 3) to the hypothesis section, so as to ensure that the contents before and after the manuscript correspond. The revisions are as follows here for your convenience.

L369-387 (Page 17 in all marked versions):

BCP facilitates the transfer of green technologies, ideas and institutions to neighboring regions (Andreopoulou 2012). Digital technologies can spread to the neighborhood through shared networks and platforms, creating a “spillover effect” that breaks the CLI (Acs et al. 2021). Also, digital technologies enable the public to access information from multiple sources and raise their environmental awareness, thus breaking the CLI (Liu et al. 2022b). The combination of digital technology and government governance has led local governments at all levels to compete to put in place digital governance regulations and systems (Raheem et al. 2020) in order to avoid being eliminated from the new environmentally oriented regional race. Therefore, if the policy is effective, other regions are bound to follow suit. In addition, regions with higher digital technology are more inclined to attract capital intervention, while regions with scarce digital technology will still retain backward productivity, thus reinforcing CLI (Ustundag, Cevikcan and Karacay 2018). This situation also exists in China. Over the past decade or so, there have been serious regional disparities in digital infrastructure (Kane et al. 2017), which in turn has been an important factor in attracting talent and capital (Barata 2019). Thus, the development of BCP can affect CLI through the “siphon effect”. In light of this, we propose the following hypothesis:

Hypothesis 3. BCP has a spatial spillover effect on CLI.

8. The Methods section does not present the full range of methods employed in the article and the role of these methods in relation to each other. While it is acceptable to detail only the main methods in this section, it is highly recommended that a figure be included to illustrate the methodology of the research.

Response 8: We would like to thank you for the valuable comment.

In response to your comments, we have drawn a clear methodological flow chart in order to increase the readability of the manuscript. In addition, we reorganize the research theory of the paper and draw the theoretical framework of BCP’s influence on CLI. We believe that your comments will greatly improve the quality of the manuscript. The revisions are as follo

---

## [Decision Letter · Decision Letter 1]

29 Aug 2024

PONE-D-24-09583R1The urban carbon unlocking effect of digital infrastructure construction: A spatial difference-in-difference analysis from ChinaPLOS ONE

Dear Dr. Chen,

Thank you for submitting your manuscript to PLOS ONE. After careful consideration, we feel that it has merit but does not fully meet PLOS ONE’s publication criteria as it currently stands. Therefore, we invite you to submit a revised version of the manuscript that addresses the points raised during the review process.

We look forward to receiving your revised manuscript.

Kind regards,

Bijay Halder

Academic Editor

PLOS ONE

Reviewers' comments:

Reviewer's Responses to Questions

**Comments to the Author**

1. If the authors have adequately addressed your comments raised in a previous round of review and you feel that this manuscript is now acceptable for publication, you may indicate that here to bypass the “Comments to the Author” section, enter your conflict of interest statement in the “Confidential to Editor” section, and submit your "Accept" recommendation.

Reviewer #1: All comments have been addressed

Reviewer #2: (No Response)

2. Is the manuscript technically sound, and do the data support the conclusions?

Reviewer #1: Yes

Reviewer #2: Yes

3. Has the statistical analysis been performed appropriately and rigorously? 

Reviewer #1: Yes

Reviewer #2: Yes

4. Have the authors made all data underlying the findings in their manuscript fully available?

Reviewer #1: Yes

Reviewer #2: Yes

5. Is the manuscript presented in an intelligible fashion and written in standard English?

Reviewer #1: Yes

Reviewer #2: Yes

6. Review Comments to the Author

**Reviewer #1:** This edition's manuscripts are better organized and presented than the previous edition. However, improvements are still needed in the language used throughout the manuscript.

The manuscript presents one significant issue that requires further attention. The study needs to clarify whether it is studying the impact of digital infrastructure on CLI or the impact of BCP on CLI.

The article's primary focus is on BCP, rather than digital infrastructure. However, the term "BCP" is not included in the title. Instead, the study's primary objective is to examine the phenomenon of carbon lock-in in the context of economic development in modern countries, with a particular focus on the role of digital infrastructure construction in addressing this issue.

Here are other issues worth considering:

1. The introduction should concentrate on the subject matter itself, i.e. on the possible link between digital infrastructure and carbon emissions. It must be presented as a general development issue, rather than as a matter solely pertinent to China. Subsequently, an introduction to China's BCP policy should be provided, accompanied by empirical research findings pertaining to the impact of BCP on CLI.

Prior to introducing China's policy, it is essential to elucidate the rationale behind the choice of BCP as a research object.

Does this indicate that China is grappling with a more significant CLI challenge?

Or is it because China's BCP policy is, to some extent, more representative and can serve as a point of reference for other countries around the world facing similar challenges?

This question should be addressed in the introduction.

2. It appears that CLI is not referenced in the introduction, which is a central concept in this study. It may be advisable for the authors to consider incorporating part of section 2.1 into the introduction. It is recommended that the introduction be shortened, as it is currently too lengthy.

3. It is recommended that section 3.2 be a section of descriptive analysis, with CLI measurement as a separate section.

Furthermore, the methodology section should elucidate how certain characteristics of BCP are integrated into the CLI measurement methodology.

4. In the conclusion, the study should return to the impact of digital infrastructure on CLI. This could be integrated into the corresponding paragraph on line 654.

**Reviewer #2:** In view of my reading of your paper, I invite you to address the issues noted below, which are relatively minor but nonetheless essential.

1. Manuscript Formatting: I strongly recommend reviewing the manuscript format to ensure it aligns with PLOS ONE's standards. This includes proper reference citation formats and consistent styling of figures and tables throughout the text.

2. Clarification of Results and Discussion Sections: There appears to be some confusion between the Results and Discussion sections. Although you have included some discussion in section 6.1 Conclusions, it lacks a clear comparison with existing literature, which should be supported by references.

Results Section: This section should objectively present your data and findings without interpretation. The language should be concise and descriptive.

Discussion Section: This section is where you should interpret the significance of your results, compare them with existing research, and explore their implications. It allows for more subjective insights and theoretical explanations.

For instance, in section 4.1 Benchmark Regression Results, both Results and Discussion are combined. I recommend reorganizing these sections to enhance the clarity and logical flow of your paper.

7. PLOS authors have the option to publish the peer review history of their article (what does this mean?). If published, this will include your full peer review and any attached files.

Reviewer #1: No

Reviewer #2: No

---

## [Author Response · Author response to Decision Letter 1]

13 Oct 2024

Response to Reviewers Comments

Dear editor and reviewers:

On behalf of my co-authors, we thank you very much for your letter and for the reviewers’ comments concerning our manuscript entitled “The urban carbon unlocking effect of digital infrastructure construction: A spatial difference-in-difference analysis from China (PONE-D-24-09583R1)”. Those comments are all valuable and very helpful for revising and improving our paper, as well as the important guiding significance to our research. We have studied the comments carefully and have made a major revision and re-plan which we hope to meet with the approval. We tried our best to improve the manuscript and all changes made in the revised manuscript are highlighted in blue that will not influence the content and the framework of the paper. The points-by-points respond to reviewers’ comments are as follows:

Reviewer # 1 comment:

1. This manuscript investigated the carbon lock-in phenomenon in the economic development of modern countries and how digital infrastructure construction addresses this issue. Taking the “Broadband China” policy as an example, this manuscript raises an interesting topic, which is of great significance to countries seeking structural transformation of their economies, especially developing countries like China.

However, it appears that the focus of this manuscript may have been misplaced. The manuscript in question examines the impact of BCP on CLI. While this issue is undoubtedly of great importance to China's economic development, such a narrow focus may diminish the academic merit of the study. This study would be more suitable for publication in a Chinese journal unless the authors decide to shift the focus of the study to reveal the impact of digital infrastructure on CLI and explore this phenomenon with China as the study area.

Response 1: Thank you for your thought-provoking suggestion. We apologize for any confusion caused by the lack of a clear research object in the title.

In this study, we focus on the impact of digital infrastructure construction on CLI and its mechanism, and take broadband China development strategy as a proxy for digital infrastructure construction. There are two main reasons for this choice: First, broadband networks are a core component of digital infrastructure, and by improving the quality and coverage of broadband networks, they provide the necessary basic support for the development, services and applications of the digital economy. Second, the broadband China strategy is an important national policy in the development of the digital economy, emphasizing the construction and upgrading of digital infrastructure. The implementation of this policy has a direct impact on the city’s digitization process and related infrastructure investment. Therefore, we believe that it is reasonable and feasible to take broadband China as a proxy of digital infrastructure construction, and this has been mentioned in many previous literatures, , and we have summarized some of the relevant literatures below.

1. Xiao, X., Liu, C., & Li, S. (2024). How the digital infrastructure construction affects urban carbon emissions—A quasi-natural experiment from the “Broadband China” policy. Science of The Total Environment, 912, 169284.

2. Li, X., Yang, G., Shao, T., Yang, D., & Liu, Z. (2024). Does digital infrastructure promote individual entrepreneurship? Evidence from a quasi-natural experiment on the “Broadband China” strategy. Technological Forecasting and Social Change, 206, 123555.

3. Peng, H. R., Ling, K., & Zhang, Y. J. (2024). The carbon emission reduction effect of digital infrastructure development: Evidence from the broadband China policy. Journal of Cleaner Production, 434, 140060.

4. Hong, J., Shi, F., & Zheng, Y. (2023). Does network infrastructure construction reduce energy intensity? Based on the “Broadband China” strategy. Technological Forecasting and Social Change, 190, 122437.

5. Zhou, X., Hu, Q., Luo, H., Hu, Z., & Wen, C. (2024). The impact of digital infrastructure on industrial ecology: Evidence from broadband China strategy. Journal of Cleaner Production, 447, 141589.

6. Li, X., He, P., Liao, H., Liu, J., & Chen, L. (2024). Does network infrastructure construction reduce urban–rural income inequality? Based on the “Broadband China” policy. Technological Forecasting and Social Change, 205, 123486.

7. Wu, W., Wang, S., Jiang, X., & Zhou, J. (2023). Regional digital infrastructure, enterprise digital transformation and entrepreneurial orientation: Empirical evidence based on the broadband China strategy. Information Processing & Management, 60(5), 103419.

8. Hu, J., Zhang, H., & Irfan, M. (2023). How does digital infrastructure construction affect low-carbon development? A multidimensional interpretation of evidence from China. Journal of cleaner production, 396, 136467.

9. Hua, Y., & Zhang, H. (2024). Labour misallocation and digital infrastructure: evidence from a quasi-natural experiment of ‘broadband China strategy’. Applied Economics Letters, 1-7.

10. Liu, Y., Liu, K., Zhang, X., & Guo, Q. (2024). Does digital infrastructure improve public Health? A quasi-natural experiment based on China's Broadband policy. Social Science & Medicine, 344, 116624.

At the same time, our main target is the phenomenon of carbon locking in the context of modern national economic development, with a special focus on the role and potential of digital infrastructure development in addressing this problem. Our research findings hope to find out the great potential of digital infrastructure construction with Broadband China strategy as a proxy to solve the carbon lock-in problem, which can provide practical experience and policy reference for other countries in the world facing similar problems. In fact, many countries have also carried out the construction of digital infrastructure similar to broadband China, such as the United States’ “National broadband Plan”, Singapore’s “smart country”, the United Kingdom’s “all-fiber network construction” digital strategy, Germany’s “broadband network” digital agenda, and India’s “Digital India” based on national fiber network upgrades.

In summary, we believe that it is very meaningful to take broadband China as a specific research object to explore the impact of digital infrastructure construction on CLI. At the same time, your comments still give us a lot of inspiration. We reorganized the title of the manuscript to avoid confusion for the readers. The revisions are as follows here for your convenience.

L1-3 (Page 1 in all marked versions):

The urban carbon unlocking effect of digital infrastructure construction: A spatial difference-in-difference analysis from “Broadband China” pilot policy

2. The introduction should concentrate on the subject matter itself, i.e. on the possible link between digital infrastructure and carbon emissions. It must be presented as a general development issue, rather than as a matter solely pertinent to China. Subsequently, an introduction to China's BCP policy should be provided, accompanied by empirical research findings pertaining to the impact of BCP on CLI.

Prior to introducing China's policy, it is essential to elucidate the rationale behind the choice of BCP as a research object.

Does this indicate that China is grappling with a more significant CLI challenge?

Or is it because China's BCP policy is, to some extent, more representative and can serve as a point of reference for other countries around the world facing similar challenges?

This question should be addressed in the introduction..

Response 2: Thank you for your kind suggestion which is of great help to the integrity of our manuscript.

Based on your comments, we have combed through the introduction and reorganized the content of this section. Specifically, we expand the introduction in the following order. First of all, we present the development context of the current social changes brought about by digitalization. Second, we review current research on the relationship between digital infrastructure development and carbon emissions. Thirdly, we summarize the development of digital infrastructure based on broadband network in other countries and explain the rationality and importance of BCP as an agent of digital infrastructure construction. Fourth, we briefly introduced the CLI. Fifth, we propose the problems and contributions to be solved by this research. The last part is the organization of the manuscript.

We believe that the reorganized introduction structure gives the manuscript a clearer logic and greatly increases its readability. The revisions are as follows here for your convenience.

L28-132 (Page 1 in all marked versions):

Currently, human society is experiencing a digital social change based on broadband Internet information, and the digital revolution brought about by the spread of information technology has become an important factor in the economic growth of each country [1]. The digital wave is driving the fourth industrial revolution in the world [2, 3]. Internet information broadband as a digital infrastructure has led to the rise of industrial Internet, cloud computing, artificial intelligence and other technologies, which in turn promotes the green transformation of traditional industries, enables different innovation bodies to share and absorb knowledge with higher efficiency and lower cost, and accelerates green technological innovation in the region through the “spillover effect” [6-9]. In addition, the digital infrastructure realizes the linkage of environmental information and resource sharing, making its role in environmental governance and supervision increasing prominent [10-12]. So, it is foreseeable that digital infrastructure provides a good opportunity for urban low-carbon transformation, industrial upgrading and green development [14, 15].

At the same time, many scholars believe that digital infrastructure has the potential to be sustainable and are optimistic about reducing carbon emissions. Romm [16] found that widespread use of the Internet resulted in a significant reduction in energy intensity in the United States and highlighted the important contribution of ICTs in reducing greenhouse gases. Moyer and Hughes [17] found that digital infrastructure could reduce global carbon emissions by about 50 years. Liu, Cong [18] verified that digital infrastructure can reduce carbon emissions in China, and there is significant regional heterogeneity in this effect. Asongu, Simplice [19] found that digital infrastructure layout has a positive impact on reducing carbon emissions in African countries. Zahra, Dehghan [20] demonstrate that ICTs can reduce carbon emissions in Iran’s transport sector and countries along the Belt and Road. On the one hand, digital infrastructure can optimize industrial structure and promote industrial transformation and upgrading. The reduction of energy use intensity in the service sector helps to reduce carbon emissions. On the other hand, digital infrastructure can promote technological progress, accelerate the research and development and diffusion of innovation, facilitate the development of clean energy, improve energy utilization efficiency, strengthen the monitoring and control of carbon emissions, and thus reduce carbon emissions. It can be seen that digital infrastructure helps to change the traditional development model from relying on large-scale energy consumption to a more low-carbon and sustainable development mode, effectively breaking the carbon lock of traditional industries, and providing key technical and structural support for the global realization of a low-carbon economic path.

Therefore, a growing number of countries are proposing plans to grow their digital infrastructure construction marked by broadband network, such as the United States, United Kingdom, and Japan [21-23]. In addition, the Chinese government has proposed a strategy known as “Information Infrastructure Construction (IIC)”. Specifically, since 2014, China’s Ministry of Industry and Information Technology has issued a “Broadband China” policy (BCP), and has approved a total of three batches of pilot cities to promote information infrastructure construction in 2015 and 2016. The policy mainly includes four contents in the upgrading of broadband users in pilot cities, namely, scale, penetration rate, access capacity and application scope. By then, China will have essentially finished building an accessible and fast internet network infrastructure.

On the one hand, BCP, as a core component of digital infrastructure, provides the necessary basic support for the development, services and applications of the digital economy by improving the quality and coverage of broadband networks. On the other hand, BCP is an important national policy for the development of the digital economy, emphasizing the construction and upgrading of digital infrastructure. The implementation of this policy has a direct impact on the city’s digitization process and related infrastructure investments. Therefore, we believe that it is reasonable and feasible to take BCP as a proxy for digital infrastructure, and it is of great significance. Our main objective is to focus on the phenomenon of CLI in the context of economic development in modern countries, with a particular focus on the role and potential of digital infrastructure development in addressing this problem. Our research results hope to find the positive impact of digital infrastructure construction represented by the Broadband network strategy to solve the CLI problem, and provide practical experience and policy reference for other countries in the world facing similar problems.

Carbon lock-in (CLI) describes a phenomenon in which the fossil fuel-based energy consumption structure under the traditional economic development model cannot be changed in the short term, thus making the economy firmly locked into a carbon-based energy system [24-26]. CLI not only impedes the advancement of low-carbon technology and fosters dependency on development pathways, but also poses a persistent danger to harmony in ecosystems and environmental conservation[27]. Therefore, there is a pressing need to find a solution to the growing CLI, thereby decoupling economic growth from traditional development models.

3. It appears that CLI is not referenced in the introduction, which is a central concept in this study. It may be advisable for the authors to consider incorporating part of section 2.1 into the introduction. It is recommended that the introduction be shortened, as it is currently too lengthy.

Response 3: We thank you for this careful comment.

According to your suggestions, we have made the following adjustments to the introduction. First of all, we added a description of the CLI in the introduction to ensure that the content before and after the manuscript is more logical. Secondly, we refined the content of the introduction, highlighted key issues, and appropriately reduced the volume of the introduction. Furthermore, we add to the rationale for BCP as a digital infrastructure building agent. Finally, we detail the goal of the study. On the one hand, China is under pressure from the dual-carbon target, and the central government is struggling to meet the challenges posed by the CLI; On the other hand, BCP is more representative of digital infrastructure, and its positive impact on economic development mode, environmental governance and climate change can provide reference and practical experience for other countries that are carrying out digital construction and facing environmental challenges and sustainable development problems.

4. It is recommended that section 3.2 be a section of descriptive analysis, with CLI measurement as a separate section.

Furthermore, the methodology section should elucidate how certain characteristics of BCP are integrated into the CLI measurement methodology.

Response 4: We would like to thank you for the valuable comment.

Based on your suggestions, we have made the following changes to this section: First, we present all the variable calculations as a separate section in Section 3.2, so as to guarantee its integrity. Second, we put the descriptive analysis of CLI in Chapter 4 results analysis (Section 4.1) to ensure 

---

## [Decision Letter · Decision Letter 2]

14 Nov 2024

PONE-D-24-09583R2The urban carbon unlocking effect of digital infrastructure construction: A spatial difference-in-difference analysis from "Broadband China" pilot policyPLOS ONE

Dear Dr. Chen,

Thank you for submitting your manuscript to PLOS ONE. After careful consideration, we feel that it has merit but does not fully meet PLOS ONE’s publication criteria as it currently stands. Therefore, we invite you to submit a revised version of the manuscript that addresses the points raised during the review process.

We look forward to receiving your revised manuscript.

Kind regards,

Bijay Halder

Academic Editor

PLOS ONE

Journal Requirements:

Reviewers' comments:

Reviewer's Responses to Questions

**Comments to the Author**

1. If the authors have adequately addressed your comments raised in a previous round of review and you feel that this manuscript is now acceptable for publication, you may indicate that here to bypass the “Comments to the Author” section, enter your conflict of interest statement in the “Confidential to Editor” section, and submit your "Accept" recommendation.

Reviewer #1: All comments have been addressed

Reviewer #2: (No Response)

2. Is the manuscript technically sound, and do the data support the conclusions?

Reviewer #1: Yes

Reviewer #2: Yes

3. Has the statistical analysis been performed appropriately and rigorously? 

Reviewer #1: Yes

Reviewer #2: Yes

4. Have the authors made all data underlying the findings in their manuscript fully available?

Reviewer #1: Yes

Reviewer #2: Yes

5. Is the manuscript presented in an intelligible fashion and written in standard English?

Reviewer #1: Yes

Reviewer #2: Yes

6. Review Comments to the Author

Reviewer #1: The authors have addressed all my questions. I believe that this manuscript has met the standards for publication.

Reviewer #2: Dear Author,

Thank you for your thoughtful revisions based on the previous review comments. After reading the revised manuscript, I have the following further suggestions:

1. Conclusions and Discussion

Currently, the boundaries between the conclusions and the discussion are not clearly defined. The discussion should be based on the research results, summarizing and interpreting them, while comparing them with existing literature, highlighting the similarities and differences, and exploring the mechanisms behind these differences. Additionally, the discussion should further address the limitations and shortcomings of the study. The conclusion should succinctly summarize the core content of the discussion, revert to the research questions, and emphasize the key findings and academic contributions. I suggest reorganizing Sections 6 and 7 to ensure logical clarity.

2. Consistency in Table Formatting

Thank you for adjusting the table formats. However, to enhance the professionalism and consistency of the paper, I recommend unifying the table style (e.g., either use three-line or full-line tables throughout).

I look forward to your revised submission and thank you for your contribution.

Best regards

7. PLOS authors have the option to publish the peer review history of their article (what does this mean?). If published, this will include your full peer review and any attached files.

Reviewer #1: No

Reviewer #2: No

---

## [Author Response · Author response to Decision Letter 2]

29 Nov 2024

Response to Reviewers Comments

Dear editor and reviewers:

On behalf of my co-authors, we thank you very much for your letter and for the reviewers’ comments concerning our manuscript entitled “The urban carbon unlocking effect of digital infrastructure construction: A spatial difference-in-difference analysis from “Broadband China” pilot policy (PONE-D-24-09583R2)”. Those comments are all valuable and very helpful for revising and improving our paper, as well as the important guiding significance to our research. We have studied the comments carefully and have made a major revision and re-plan which we hope to meet with the approval. We tried our best to improve the manuscript, and all changes made in the revised manuscript are highlighted in blue that will not influence the content and the framework of the paper. The points-by-points respond to reviewers’ comments are as follows:

Reviewer # 2 comments:

1. Currently, the boundaries between the conclusions and the discussion are not clearly defined. The discussion should be based on the research results, summarizing and interpreting them, while comparing them with existing literature, highlighting the similarities and differences, and exploring the mechanisms behind these differences. Additionally, the discussion should further address the limitations and shortcomings of the study. The conclusion should succinctly summarize the core content of the discussion, revert to the research questions, and emphasize the key findings and academic contributions. I suggest reorganizing Sections 6 and 7 to ensure logical clarity.

Response 1: Thank you for your thought-provoking suggestion. Based on your comment, we have reorganized the content of sections 6 and 7 to ensure that the logic is clearer. The revisions are as follows here for your convenience.

L587-650 (Page 39 in all marked versions):

6.1 Discussion

Our key findings suggest that BCP make a significant contribution to breaking the CLI, which is consistent with existing literature supporting digital infrastructure as a way to reduce carbon emissions. Digital infrastructure drives green technology innovation, improves resource efficiency while reducing environmental pollution. More importantly, it brings about a change in the way the city develops, altering its development model that is overly reliant on traditional resources, thus breaking the CLI [36, 39, 43, 45]. When the traditional development model continues, the output performance of high pollution, high emission and low efficiency will further aggravate CLI and even affect climate change [21, 22]. Therefore, reducing CLI should shift from an economic model with high energy consumption and high carbon emissions to a low-carbon circular development model with high efficiency and high output. Fortunately, this shift can be realized through the digital development strategy.

At the same time, BCP accelerates technological innovation and industrial structure upgrading, thus contributing to carbon unlocking. This research adds well to existing literature. In the extensive discussion of the influential factors of CLI, the positive effects of digitalization have been overlooked in the literature [27, 33, 36-39]. These findings provide new perspectives by revealing how digital infrastructure can break the CLI through technological innovation and industrial structure upgrading. The advancement of digital technology has been an important catalyst for technological innovation, industrial transformation and consumption upgrading [3, 12, 46, 82].

More importantly, on the one hand, our findings verify the importance of digital infrastructure in the coordinated development of regional economies, that is, it is crucial in promoting the coordination of inter-regional environmental governance; on the other hand, the policy lag effect found in this study provides an important time reference dimension for policymakers, and also emphasizes the need to consider medium- and long-term impacts when evaluating the implementation effect of such programs. However, the above two key issues have never been discussed in previous studies.

6.2 Conclusion

Based on an empirical analysis of 266 prefecture-level cities in China from 2006 to 2019, this study explores the effects, mechanisms, synergies and heterogeneity of the BCP on CLI. The results are as follows:

(1) BCP can significantly reduce CLI, and after a number of robust tests, the finding is still valid.

(2) The impact of BCP on CLI has a spatial spillover effect, that is, BCP can contribute to CLI in both neighboring and local cities.

(3) There is a lag in the policy effect of BCP, i.e., its inhibitory effect on CLI becomes significant only from the third year of policy execution.

(4) The contribution of BCP to CLI is more significant in the eastern region, resource-based cities, and cities with a better digital base.

(5) BCP is able to break CLI by promoting green technological innovation and facilitating industrial structure upgrading.

With the increasing severity of the global climate change problem and the pursuit of sustainable development, achieving carbon neutrality has become a common challenge faced by various countries. As a policy initiative covering economic, social and environmental dimensions, the impact of BCP on carbon emissions is not only related to the realization of China’s future carbon emission reduction targets, but also provides empirical support and policy reference for global carbon management and climate change governance. By systematically studying the carbon locking effect of BCP, we can not only provide theoretical support and practical guidance for the formulation and implementation of similar policies worldwide, but also further address global carbon management, so as to solve the environmental challenges brought by climate change and promote the development and realization of sustainable development goals in other countries and regions.

2. Thank you for adjusting the table formats. However, to enhance the professionalism and consistency of the paper, I recommend unifying the table style (e.g., either use three-line or full-line tables throughout).

Response 2: Thank you for your kind suggestion. Based on your comment, we have adjusted the format of the form to ensure professionalism and consistency. We used the three-line tabular format to ensure the consistency of the tables in the manuscript. Please refer to the manuscript with all marked versions for detailed revisions.

We sincerely thank you for your professional review work, constructive comments, and valuable suggestions on our manuscript.

---

## [Editor Report · Decision Letter 3]

8 Dec 2024

The urban carbon unlocking effect of digital infrastructure construction: A spatial difference-in-difference analysis from "Broadband China" pilot policy

PONE-D-24-09583R3

Dear Dr. Chen,

We’re pleased to inform you that your manuscript has been judged scientifically suitable for publication and will be formally accepted for publication once it meets all outstanding technical requirements.

Kind regards,

Bijay Halder

Academic Editor

PLOS ONE
---

## [Editor Report · Acceptance letter]

26 Dec 2024

PONE-D-24-09583R3 

PLOS ONE

Dear Dr. Chen, 

I'm pleased to inform you that your manuscript has been deemed suitable for publication in PLOS ONE. Congratulations! Your manuscript is now being handed over to our production team.

Kind regards, 

on behalf of

Mr. Bijay Halder 

Academic Editor

PLOS ONE